# Spatiotemporal patterns, triggers and anatomies of seismically detected rockfalls

Michael Dietze[1], Jens M. Turowski[1], Kristen L. Cook[1], and Niels Hovius[1]

[1]GFZ German Research Centre for Geosciences, Section 5.1 Geomorphology, Potsdam, Germany

*Correspondence to:* Michael Dietze (mdietze@gfz-potsdam.de)

**Abstract.**

Rockfalls are a ubiquitous geomorphic process and natural hazard in steep landscapes across the globe. Seismic monitoring can provide precise information on the timing, location and event anatomy of rockfalls, parameters that are otherwise hard to constrain. By pairing data from 49 seismically detected rockfalls in the Lauterbrunnen Valley, Swiss Alps, with auxiliary meteorologic and seismic data of potential triggers during autumn 2014 and spring 2015, we are able to i) analyse the evolution of single rockfalls and their common properties, ii) identify spatial changes in activity hotspots, iii) and explore temporal activity patterns at different scales, ranging from months to minutes, to quantify relevant trigger mechanisms. Seismic data allows the classification of rockfall activity into two distinct phenomenological types. The signals can be used to discern multiple rock mass releases from the same spot, identify rockfalls that trigger further rockfalls, and resolve modes of subsequent talus slope activity. In contrast to findings based on discontinuos methods with integration times of several months, rockfall in the monitored limestone cliff is not spatially uniform but shows a systematic downward shift of a rock mass release zone following an exponential law, most likely driven by a continuously lowering water table. Freeze-thaw-transitions, approximated at first order from air temperature time series, account for only 5 out of the 49 rockfalls whereas 19 rockfalls were triggered by rainfall events, with a peak lag time of 1 h. Another 17 rockfalls were triggered by diurnal temperature changes and occurred during the coldest hours of the day as well as during the highest temperature change rates. This study is thus the first one to show direct links between proposed rockfall triggers and the spatio-temporal distribution of rockfalls under natural conditions, and extends existing models by providing seismic observations of the rockfall process prior to the first rock mass impacts.

## 1   Introduction

Rockfall is a fundamental geomorphic process in alpine landscape dynamics and an important natural hazard. Knowing where, when and due to which triggering mechanisms rockfalls occur and how they evolve are essential questions across scientific disciplines. However, rockfalls involve the infrequent and rapid mobilisation of comparably small volumes of rock, which are difficult to observe directly. As a consequence, precise constraints on timing, location and triggers are hard to come by. There are many established approaches to detect rockfall activity spatially, for example surveys of talus slopes, dendrometric and lichenometric approaches (Matsuoka and Sakai, 1999; Stoffel et al., 2005; Krautblatter et al., 2012), and more recently image-based mapping as well as terrestrial and airborne laser scanning (Stock et al., 2011; Strunden et al., 2014; D'Amato

et al., 2016). The temporal information delivered by these methods is not very precise as it is bound to the survey lapse times, which are typically on the order of weeks to years. D'Amato et al. (2016) were able to narrow the temporal resolution to the sub-daily level during a study of a limestone cliff. They analysed 10 minute interval photo imagery together with terrestrial laser scan data for a period of 887 days, which resulted in a data base of 144 rockfalls with a time uncertainty of less than 20

h for some of the events. However, in general, it has so far been difficult to link detected rockfall events to potential trigger mechanisms by temporal coincidence, and to investigate potential early warning signals at or below hourly resolution.

Seismic sensors provide a valuable complement to the above mentioned methods. They allow precise temporal fixes of rockfall event initiation and duration because they record continous high resolution signals of geomorphic activity. If the sensors are deployed as a seismic network, they further allow source location estimates with uncertainties of tens of metres (Lacroix

and Helmstetter, 2011; Burtin et al., 2014; Hibert et al., 2014; Dietze et al., 2017), enabling direct temporal and spatial links to potential triggers. Furthermore, seismic signals allow insight into the anatomy of geomorphic processes through interpretation of the recorded time series and spectral properties. During the last decade, there has been significant progress in theory (Gimbert et al., 2014; Larose et al., 2015), experiments (Vilajosana et al., 2008; Farin et al., 2015), and application across different scales (Dammeier et al., 2011; Lacroix and Helmstetter, 2011; Zimmer et al., 2012; Ekström and Stark, 2013; Burtin et al., 2016).

However, rockfall activity has mainly received attention as "by-product" of seismic observatories with different scopes (e.g., Dammeier et al., 2011; Hibert et al., 2011; Burtin et al., 2014) and research has focused on linking seismic properties with geometric and kinetic characteristics, such as mobilised volume, run out length or fragmentation (Dammeier et al., 2011; Ekström and Stark, 2013; Hibert et al., 2014). Systematic linking of events to more than one potential environmental trigger has received only marginal attention (Helmstetter and Garambois, 2010; Burtin et al., 2013; D'Amato et al., 2016).

We employ environmental seismology, the study of the seismic signals emitted by Earth surface processes, in the Lauterbrunnen Valley, a steep cliff in the Bernese Oberland prone to small rockfalls, usually below 1 m$^3$ (Strunden et al., 2014). Based on a previous study by Dietze et al. (2017) that focused on assessing validity, precision and limitations of the seismic approach to detect and locate rockfalls during an about one month long control period with auxiliary TLS data, we now investigate a longer measurement period, beyond the TLS-based control data, and use the full range of information available through

environmental seismology. We detect and locate rockfalls over a period lasting more than six months and interpret the seismic data to gain insight into the individual stages and overall phenomenological types of rockfalls, in addition to building an inital event catalogue. Based on rockfall event lag times to auxiliary environmental data, we develop a framework for parameterising and evaluating the significance of different trigger mechanisms. Combining spatial and temporal rockfall patterns, we identify a rockfall activity zone that consistently shifts down the cliff over the course of the season, and quantify the effect of diurnal

forcing on event activity within the composed catalogue.

## 2   Rockfall triggers

Rockfall is the result of the segregation of a volume of rock from the source rock mass (block production phase) and its subsequent detachment by a release mechanism activated by a driving force (trigger phase). Block production can be attributed

to several processes, such as crack propagation and dissolution of solids (Wieczorek, 1996; Krautblatter et al., 2012; Stock et al., 2013), and usually acts over several months to millions of years. The release mechanism essentially causes a decrease in the stabilising forces and/or an increase in stress until material fails and the rock mass is mobilised. We broadly follow Wieczorek (1996) in defining a trigger as an external stimulus that causes a near-immediate geomorphic response by decreasing material strength or increasing stress. Implicit to this definition is that some triggers have nearly immediate response while others require a certain response time or minimum cumulative impact duration. Some of the trigger mechanisms can also contribute to block production. However, this role is not discussed here.

Rockfall triggers are numerous and hard to assign to specific events (Stock et al., 2013). The relationship between cause (trigger) and effect (rockfall) is predominantly constrained based on temporal coincidence with almost no or only very generalised information on spatial coincidence. Trigger mechanisms can overlap, be superimposed, or have additive effects, which can complicate associating individual processes based on only response time lags. Thus, addressing the timing of cause and effect as precisely as possible is an essential precondition for resolving these effects. The following description of rockfall trigger mechanisms (cf. tables 1,2) and their anticipated effects builds the conceptual foundation for relating rockfall events identified in this study to external stimuli, i.e., to conduct a posterior process-response analysis.

## 2.1 Geophysical triggers

Earthquakes, as well as volcanic tremors and eruptive activities, generate seismic waves that result in ground acceleration and thus mechanical stress through inertial forces (Hibert et al., 2014). When this force overcomes a given threshold (e.g., set by friction or cohesion force), the rock mass can be mobilised. A typical proxy for geophysical trigger intensity is peak ground acceleration. The reaction of a rock mass to excitation by an earthquake is almost immediate, i.e., during or within seconds after the trigger, in contrast to land slides, where a time lag can be more likely (e.g., Lacroix et al., 2015).

## 2.2 Mass wasting processes

Snow avalanches can dislodge and entrain loose rocks by direct impacts or basal shear stress (Stock et al., 2013). However, snow avalanches rarely occur on cliff faces because these are too steep to support massive continuous accumulations of snow. In such terrain, ice falls are more likely to occur wherever frozen waterfalls exist. Slope failures can also be caused by destabilisation and ground motion induced by other mass movements or fluvial activity as has been shown for debris flows and rock avalanches in the Illgraben, a steep catchment in the Rhone Valley (Burtin et al., 2014). The response of a rock mass to the trigger role of other mass wasting processes is presumed to be immediate.

## 2.3 Meteorological triggers

Precipitation, particularly in the form of rain and subsequent run off, can affect rockfall activity through several mechanisms. It can increase the weight/load of a rock volume, increase pore pressure and thus decrease cohesion, lead to expansion of clay minerals, erode cohesive fine material from cracks and dissolve rock compounds (Stock et al., 2013). The reaction time

**Table 1.** Summary of rockfall triggers and potential approaches to survey/monitor them. Seismic approaches and their references were chosen with based on existing links of trigger investigations and mass wasting processes.

| Domain | Trigger | Mechanism | Lag time | Traditional survey | Seismic approach | Example reference |
|---|---|---|---|---|---|---|
| Geophysical | Earthquake | Ground acceleration | immediate | seismic monitoring | global or local network signal interpretation | (Dietze et al., 2017) |
| | Volcanic activity | Ground acceleration | immediate | seismic monitoring | local network signal interpretation | (Hibert et al., 2014) |
| Mass-wasting | Snow or rock avalanches | Impact, basal shear stress | immediate | Remote sensing, infra sound | single station, seismic antenna, catchment-wide network signal interpretation | (Suriñach et al., 2005) |
| | Ice or rock falls | Impact | immediate | Remote sensing, mapping | single station, seismic antenna, catchment-wide network signal interpretation | (Helmstetter and Garambois, 2010; Dietze et al., 2017) |
| | Debris flows | Undercutting, ground acceleration | immediate | Mapping | single station, seismic antenna, catchment-wide network signal interpretation | (Burtin et al., 2014) |
| Meteorological | Rain | Loading of the rock mass | hours | weather station | single station signal interpretation | (Turowski et al., 2016) |
| | | Increase of pore pressure | hours | point or line measurements | coda wave interferometry | (Larose et al., 2015) |
| | | Expansion of clay minerals | hours | point measurements | n.a. | n.a. |
| | | Erosion and dissolution | hours | point measurements | n.a. | n.a. |
| | Wind | Pressure fluctuations | immediate | weather station | single station signal interpretation | (Lott et al., 2017) |
| | | Leverage effects | immediate | accelerometers | single station signal interpretation | (Dietze et al., 2015) |
| | Lightning strike | Gas pressure increase | immediate | Electromagnetic pulse or radio frequency detector networks | single station or local network signal interpretation | (Kappus and Vernon, 1991) |

of a rock mass to precipitation depends on the exact mechanism. Increasing the load beyond the water film adhering to or running over the surface of a rock mass requires time for rain water infiltration, percolation and retention inside the rock mass. Thus, rainfall amount and surface permeability are further important control factors. Pore pressure decrease also occurs after

**Table 2.** Summary of rockfall triggers and potential approaches to survey/monitor them. Seismic approaches and their references were chosen with based on existing links of trigger investigations and mass wasting processes.

| Domain | Trigger | Mechanism | Lag time | Traditional survey | Seismic approach | Example reference |
|---|---|---|---|---|---|---|
| Heat-related | Thaw-freeze | Pressure increase by volume expansion | immediate to minutes | point measurements | coda wave interferometry | (Larose et al., 2015) |
| | Freeze-thaw | Cohesion loss | immediate to minutes | laboratory experiments | coda wave interferometry | (Larose et al., 2015) |
| | | Stress field reorganisation | immediate to minutes | laboratory experiments | coda wave interferometry | (Larose et al., 2015) |
| | | Additional melt water production | minutes to hours | point or line measurements | coda wave interferometry | (Larose et al., 2015) |
| | | Ice volume expansion below thawing point | minutes to hours | point or line measurements | coda wave interferometry | (Larose et al., 2015) |
| | Thermal gradients | Stress due to dilation and contraction | hours | point measurements | coda wave interferometry | (Larose et al., 2015) |
| | | Crack propagation | hours | point measurements | coda wave interferometry | (Larose et al., 2015) |
| | | Ratchet mechanism | hours | point measurements | n.a. | n.a. |
| Biological and anthropogenic | animal/human traffic | ground vibrations, dislodgement | immediate | video imagery | single station signal interpretation | n.a. |
| | vegetation growth | growing load | n.a. | time lapse imagery, mapping | n.a. | n.a. |
| | | leverage effects through wind | immediate | vegetation instrumentation | single station signal interpretation | (Dietze et al., 2015) |
| | human activities | ground motion due to diverse sources | immediate | diverse monitoring techniques | single station or local network signal interpretation | n.a. |

percolation until, eventually, the entire regolith or rock mass is saturated. Both processes may show lag times of several hours, depending on the local hydrology. Even longer lag times are to be expected for swell-shrink effects of clay minerals. Most of the mentioned triggers change the material structure of the rock mass to fail and could in principal be investigated by coda wave interferometry methods (see Larose et al. (2015) for a review of possible techniques).

Wind interaction with bare rock surfaces results in pressure fluctuations and thus cyclic stress. Trees or other perennial plants can cause a local leverage effect, especially when their roots have penetrated into cracks and fissures of a rock mass (see also section 2.5). The response of a rock mass to excitation by wind should be immediate as there is no mechanism that would cause a time lag.

Lightning can contribute to rock fracturing and mobilisation by the massive electric discharge that is able to vaporise water and thus increase gas pressure within the rock. There has been speculation about the role of lightning in the erosion of mountain summits (Knight and Grab, 2014). There should be no time lag between lightning strike and rockfall activation.

## 2.4 Heat-related triggers

This group includes two mechanisms: freeze-thaw dynamics and thermal stress. Freeze-thaw actions as rockfall triggers can work in two directions: transitions from the liquid to the solid state and vice versa (D'Amato et al., 2016, and references therein). During freezing, volume expansion through ice formation drastically increases rock-internal pressure but also increases the cohesion along rock joints, which has a positive effect on bulk rock strength. During thawing, the stress field created by the interplay between rock structure and ice-filled cracks and fissures changes suddenly. Additional melt water is produced, with consequences for the rock mass similar to those of water from precipitation. A further process is the warming of ice below the melting point, which can cause pressure to increase due to thermal dilation. D'Amato et al. (2016) find the most common response times at their highest resolution level ($< 20$ h), which means that slope reaction to freeze-thaw transitions can be expected to be 20 hours or less.

Thermal stress results from rock deformation due to heat-driven dilation or contraction. This cyclic mechanism can prepare blocks by driving crack propagation and finally cause the failure itself (Collins and Stock, 2016). Furthermore, material that falls or is washed into the opened fissures prevents the fissure from closing again and thus further increases stress (Ratchet mechanism, Bakun-Mazor et al. (2013)). There are two parameters of interest: the extreme states of deformation (maximum contraction and maximum expansion) and the deformation rate. The time lag between thermal forcing and rockfall results primarily from heat diffusion into the rock mass and the subsequent deformation. Calculations and in situ measurements by Collins and Stock (2016) for heating a 10 cm thick granitic rock slab in a rockfall prone environment by 20 K suggest a diffusion time of about 3 hours. Thus, for propagating a heat pulse through the first few decimetres of rock, time lags of several hours can be expected.

Like the rain-related triggers, heat-related mechanisms also change the material structure significantly and may thus be investigated by coda wave interfreometry approaches (Larose et al., 2015). However, this promising scientific field with its application to Earth surface process research topics has just emerged recently and is still at an experimental stage.

## 2.5 Biotic/anthropogenic triggers

Biological triggering of rockfalls can be through animal traffic on loose rocks as well as vegetation growth. The latter results in a growing load with time, leaching of mineral components, and also the expansion of rock fractures by the root system and leverage effects through interaction with wind. We are not aware of a study that explicitly links the effects of biological activity to rockfall activity. Thus, a time lag discussion for this trigger would be highly speculative.

Human activity is manifold. It can cause rockfall by ground vibrations due to transport activity such as train or road traffic, construction work (including resulting terrain disturbance), and blasting, as well as direct rock dislodgement by people passing on foot or climbing. The response of rockfall to this trigger mechanism is supposed to be immediate.

Except for increased load due to vegetation growth, these biological trigger mechanisms can be sensed seismically. However, depending on the intensity of the signals, the seismic sensor must be at close distance to the source, which requires a dense network of stations with apertures of not more than a few kilometres.

## 3 Materials and methods

### 3.1 Study area

The Lauterbrunnen Valley (figure 1) is a spectacular Alpine valley with about 1000 m high, nearly vertical (88.5 °) Mesozoic limestone cliffs (part of the Doldenhorn Nappe, predominantly massive rock but showing both brittle and ductile deformation (BfL, 2004)), which are dissected by several hanging valleys. About 150 m high talus slopes at the base of the cliff, in many locations covered with fresh debris, suggest substantial and sustained rockfall. In winter, the rock wall is snow-free, but the waterfalls usually freeze. The weather in Mürren, on top of the cliff at about 1630 m asl. is humid with precipitation amounts of 1554 mm per year and a temperature range of -4 to 12 ° C. The steepest section of this rockfall-prone valley, located between the towns of Mürren and Lauterbrunnen, ranging between 1600 and 800 m asl., has been investigated in an earlier study, where terrestrial laser scan data was combined with seismic data (Dietze et al., 2017). This combination of methods allowed pairwise detection of ten rockfall events, ranging from $0.053 \pm 0.004$ to $2.338 \pm 0.085$ m$^3$ within one month, with location differences of $81^{+59}_{-29}$ m. Thus, for this area (about 2.16 km$^2$) the data processing work flow and validation of the seismic approach has already been developed. Under the current conditions, rockfall activity mobilises small volumes, usually below 1 m$^3$, and appears to be more or less equally distributed throughout the monitored cliff faces when integrated over several months (Strunden et al., 2014). In contrast, when determining event timing and location at sub-diurnal intervals (Dietze et al., 2017), events are highly episodic and spatially non-uniformly distributed.

### 3.2 Equipment and deployment

Seismic activity was monitored by six broadband seismometers (Nanometrics Trillium Compact 120s). The instruments were deployed during two observation periods: 30 July to 28 October 2014 and 17 March to 24 June 2015. Ground velocity signals were recorded with Omnirecs Cube ext[3] data loggers, sampling at 200 Hz (gain of 1, GPS flush time 30 minutes). Deployment

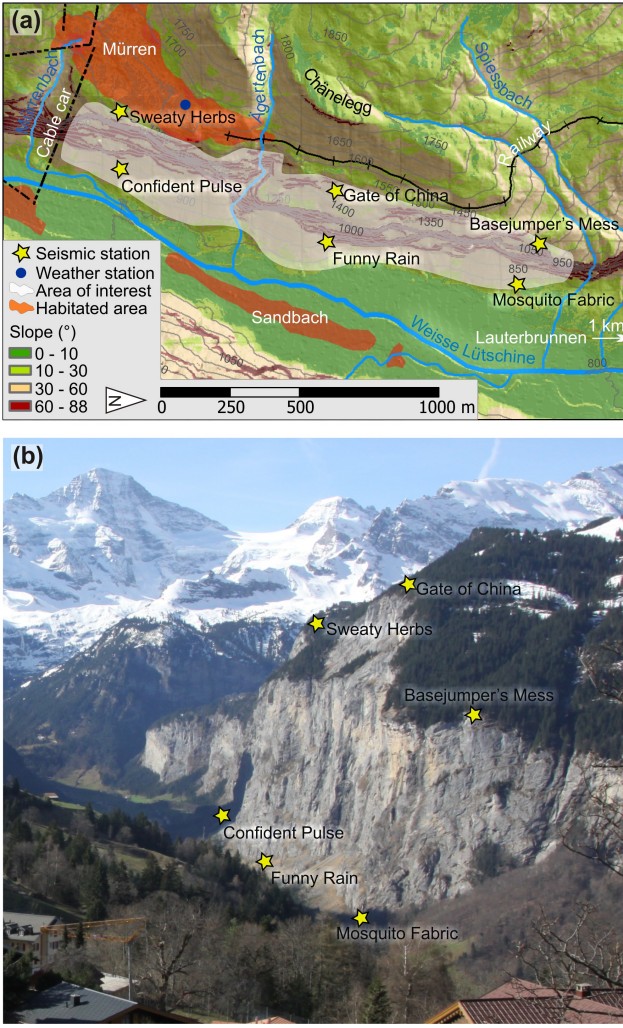

**Figure 1.** The study area Lauterbrunnen Valley. a: Schematic map with location of seismic stations and weather station as well as anthropogenic noise sources (settlements, technical infrastructure). b: Photograph of the instrumented east-facing rock wall.

sites were chosen to optimise the potential for event location along the east-facing rock wall below the town of Mürren. Stations were separated from each other laterally by 1100 to 1300 m and vertically by 700 to 1000 m. Three stations were deployed along the upper limits of the talus slopes at the cliff base and three stations were set up on top of the cliff (cf. figure 1). Each seismic sensor was installed in a small hand dug pit at 30 to 40 cm depth, seated on bedrock where possible.

5    For locating the seismic sources due to rockfall, a digital elevation model (DEM) of the wider study area with 5 m grid size (swissALTI3D) was projected to the UTM system and resampled to 10 m grid size to decrease computational time during the location approach. For quality assessment and source location projection along the vertical cliff, a high resolution topographic model of the valley wall was created using terrestrial lidar data collected with a Riegl VZ-6000 scanner in March 2014. Scans

collected from four different vantage points were combined to cover the full cliff face over a horizontal distance of about 6 km with point spacing between 0.2 and 0.5 m. The resulting point cloud was subsampled to obtain a resolution of 1 m and was georeferenced using the 5 m swissALTI3D DEM. Hourly data of air temperature, precipitation and global radiation from a weather station in Mürren (cf. figure 1, data from Meteomedia) were analysed to relate the identified rockfall events to
meteorological triggers.

## 3.3  Seismic data analysis

### 3.3.1  Detection

Detection of rockfall events was performed with the same approach and parameter settings as in a previous study of the same processes in this study area (cf. Dietze et al., 2017, for a justification and discussion of parameters). The vertical component
signal of the central station, "Gate of China", was screened for seismic events using a short-term-average/long-term-average (STA/LTA) ratio picker (Allen, 1982). This algorithm is sensitive to instantaneous rises in the recorded seismic signals, which affect the long-term running average only marginally while raising the short-term running average severely and thus increase the ratio of the two at the onset of seismic activity. The STA-LTA picker can be used to define the start (on-threshold STA-LTA ratio) and the end of an event (off-threshold STA-LTA ratio), i.e., to extract discrete events from the continuous stream of
seismic data. For this, the hourly raw signal files from both monitoring campaigns were collated into daily traces with a one hour overlap. These time series were filtered between 10 and 30 Hz, the typical frequency band of rockfalls and rock avalanches (Helmstetter and Garambois, 2010; Hibert et al., 2014; Burtin et al., 2016), and their signal envelopes (i.e., the square root of the squared Hilbert transform of the signal) were calculated. The STA/LTA picker was run with a short term window of 0.5 s and a long term window of 90 s, to be senstive to short pulses in the signals (cf. Burtin et al., 2014). The on-threshold was set
to 5, the off-threshold was set to 2, a combination that yielded the optimal compromise between valid detection of small events and false alarms. The long-term average value was set constant after the start of an event (cf. Burtin et al., 2014). Following Dietze et al. (2017), events that were longer than 20 s (typically earthquakes) or were shorter than 0.5 s (typically local raindrop impacts) were removed. The 20 s constraint was only applied for the STA/LTA picking step. This means, rock falls would only by rejected if all of their subsequent impact signals would last more than 20 s each. Likewise, events with a signal-to-noise-ratio
(SNR, defined as ratio of maximum to mean signal amplitude of a picked event) below 6 were removed as they could not be safely interpreted as target events (see below) and appropriately locating them would have been problematic. Further events were excluded when the time delay with which their signal arrived at the seismic stations was higher than the time a seismic wave would need to travel through the array, which was 1.4 s for the average apparent velocity in this area (Dietze et al., 2017). Location of rockfalls is only meaningful when the same seismic source (e.g., detachment process or impact) is recorded by
the stations. Allowing for larger time windows would indeed cause triggering of different event phases by different stations and consequently, at least a smearing of the location estimate. Thus, the STA/LTA results of all other stations for the picked event with a two-sided buffer of 1.4 s were checked for coincidence. Only when an event was detected by at least three seismic stations the result was kept. When two or more events were identified within less than 12 s (the maximum free fall duration

from the top of the cliff), only the first one was kept and the others were treated as potentially successive impacts of the rock mass at lower cliff sections. The goal of this restrictive signal processing approach is to effectively remove false alarms in a seismically noisy environment while detecting weak signals caused by small rockfalls. The goal is not to automatically detect the correct event onset and end timing. These information are gathered by manually inspecting the remaining valid signals.

### 3.3.2 Description of signals

Ten rockfall events in the Lauterbrunnen Valley that were detected by laser scanning as well as seismic methods (Dietze et al., 2017) had seismic characteristics that are different from those of previously published rockfalls or rock avalanches in less steep terrain, which usually show emergent waveforms with slow rising and falling seismic signal time series (Helmstetter and Garambois, 2010; Hibert et al., 2014). In the Lauterbrunnen Valley, the released rock mass typically experiences a significant free fall phase, followed by a powerful impact either somewhere on the cliff or directly on the talus slope. The impact may result in fragmentation of the initial rock mass and/or mobilisation of detritus on the talus slope. Thus, rockfalls in the study area have a distinct seismic signature in comparison to earthquakes and other mass wasting processes, such as debris flows or landslides (e.g., Burtin et al., 2016). All remaining potential rockfall events were manually checked for distinctiveness from signals of these other mass wasting and tectonic events as well as potential anthropogenic signals.

In addition to the waveforms of potential events, their spectral evolution with time was investigated using power spectral density estimates (PSD, or spectrograms). These were calculated for the event duration plus a two-sided buffer of 30 s, using multi taper correction of the spectra and the method of Welch (1967). The spectra were calculated from the deconvolved and filtered vertical component of the signal (1–90 Hz) at the central station along the cliff base ("Funny Rain") with time windows of 1.1 and 1.5 s and overlaps of 90 %. Rockfall impacts appear as sharp pulses of seismic energy over a wide frequency band, usually between 5 and 60 Hz (Lacroix and Helmstetter, 2011; Hibert et al., 2014). Rock avalanches show an emergent onset dominated by low frequencies, progressively increasing higher frequency content until the event ends with the prevalence of low frequencies (Suriñach et al., 2005). Earthquakes show the dominance of frequencies below 5 Hz and either two distinct wave train arrival times followed by an exponentially decreasing tail (coda) or a very low frequency waveform (teleseismic events). Anthropogenic signals can take a range of forms in this area (cf. Dietze et al., 2017).

### 3.3.3 Location of events

For all manually confirmed rockfall signals the source location was estimated by the signal migration method (Burtin et al., 2014). This approach is based on finding the location with the highest joined cross correlation of signal envelopes from all station pairs with time offsets. These time offsets correspond to the finite travel time of the signal from a potential source along the surface or through bedrock to each seismic station. For this, the average seismic wave velocity was set to 2700 m s$^{-1}$, which provided the best location accuracy in this study area (Dietze et al., 2017). The input signals were clipped to their STA/LTA-based start and duration plus a two-sided buffer of 2 s unless manual modification was necessary, e.g., when obviously unrelated seismic signals like rain drop impacts at one station had biased the process or when two consecutive impacts had been included. The clipped signals were filtered with four different initial cut off frequencies and the location result with the highest joined

cross correlation value was kept. These initial frequency windows were 5–15 Hz, 10–20 Hz, 10–30 Hz and 20–40 Hz. When an event could not be located in the area of interest (fig. 1) but showed all characteristics of a valid rockfall event, the frequency windows were adjusted according to the dominant frequency range. The migration operations resulted in grids with joined cross correlation values for each pixel, which may be interpreted as a probability estimate of the most likely location of the

impact that causes the seismic signal. In accordance with the findings of Dietze et al. (2017) only pixels with cross correlation values above the quantile 0.97 were kept, as this threshold resulted in the smallest possible location estimate area that still included all ten control events. The resulting data sets were normalised between 0 and 1 to have a common base for further analyses.

## 3.4  External trigger analysis

All identified rockfall events were put into context with potential trigger mechanisms by calculating the lag time to the closest preceding occurrence of each potential trigger. Depending on the mechanism, automatic algorithms or manual checks were necessary, as explained below. Interconnection and superposition of trigger mechanisms was accounted for by checking if any of the events showed a meaningful process-relevant lag time for more than one trigger.

Lag time distribution patterns for all detected rockfalls were inspected by kernel density estimates (KDE, i.e., curves that

describe the distribution of discrete empiric data). It is known that the size of the kernel (i.e., the window that is moved over the sample distribution to create the density estimate) has significant impact on the resulting curves, especially for small sample sizes (Galbraith and Roberts, 2012; Dietze et al., 2016) and there is no general rule to find the best setting. To account for this effect as well as to check the general robustness of the temporal patterns, multiple KDE graphs were generated based on Markov Chain Monte Carlo methods. For each test, the complete data set of rockfall events was subsampled 1001 times

with a random sample size between 80 and 100 % and randomly assigned kernel bandwidths. This resulted in 1001 possible realisations of density estimate curves, which were all plotted over each other to create a "ghost graph" (Blaauw, 2012) that gives a direct impression of the uncertainty associated with this method. Initial tests showed that stable, reproducable plots emerge with already 500 MCM runs and larger chains did not improve the quality of the results.

### 3.4.1  Excluded triggers

The setting of the Lauterbrunnen Valley already allows for the elimination of some of the trigger mechanisms summarised in section 2. Volcanic tremors and eruptive activities are very unlikely to influence this region of the Alps, as the nearest active volcano is Vesuvio. Snow melt-generated water input into the cliff face is regarded as irrelevant because the cliff face is snow free in winter due to the steep gradient. Only the small ledges may support accumulations of snow that could supply local input of melt water. Snow melt may be a significant source of water input in the upper parts of the catchment, above the cliff face,

but this run off would already be channelised in the hanging valleys by the time it reaches the cliff. It can thus be neglected as a mechanism significantly affecting material properties outside the hanging valleys. Likewise, snow avalanches are unlikely to influence rockfall activity along the cliff face. Root penetration of trees is considered to be of minimum relevance given the steepness of the cliff; trees only grow on the flat parts of the large ledges in the central upper section and at the southern margin

of the instrumented cliff section. Thus, these trigger mechanisms are not considered further in the article, reducing the analysis to the following mechanisms.

### 3.4.2 Geophysical triggers

Earthquakes were picked from the signals recorded by the seismic sensors with the STA/LTA approach using a 1 s short term window, a 90 s long term window as well as on- and off-ratios of 4 and 2, respectively. These parameters successfully picked all earthquakes from a ten day long control period at the beginning of the monitoring data set. The protocol was applied to data from station "Gate of China", filtered between 1 and 5 Hz, with a minimum event duration of 3 s. All picked events were checked manually for plausibility. Furthermore, the online portal of the Swiss Seismological Survey (Service) was queried for any earthquake above Mw 1 that occurred within a radius of 20 km around the study area. These values were chosen conservatively based on critical magnitude estimates for landslides and rockfalls in road cuts (Meunier et al., 2007; Jibson, 2011; Marc et al., 2016). The online catalogue contains several of the earthquakes picked by the stations used in this study, but may miss local earthquakes that could act as rockfall triggers.

### 3.4.3 Meteorological triggers

Lag times for precipitation were defined as the time span between the end of a precipitation event with $> 0.1$ mm h$^{-1}$ (the smallest increment of the meteorological data set) and the next rockfall. For each of these precipitation events the cumulative precipitation amount was calculated by backward summation in time until the beginning of the precipitation event.

Wind was investigated as a trigger using the meteorological time series data. Wind speed values were selected for the hour during which a rockfall occurred. Since there is no meaningful way to objectively determine a threshold of minimum wind speed that could serve as a trigger, a different approach was chosen. Effectiveness of wind was tested by comparing the wind speed distribution function from hours during which rockfalls occurred with 1001 randomly generated distribution functions for the entire monitoring period. If the wind speed regime during a rockfall is different from random regimes, this would be visible from this comparison. Since wind is assumed to be a regional phenomenon, a point measurement of wind speed at the station in Mürren is assumed to be representative, being aware of the drawback that we rely only on one station and that the elevation gradient of several hundred metres may result in a spatially non-consistent action of wind. However, this effect would only be relevant for particularly strong wind episodes.

For the effect of lightning there is no independent record. However, thunder also generates a seismic signal. The frequency spectrum of such a thunder signal is similar to quarry blasts, and is very broad (above 5 Hz) with peak frequencies between 6 and 13 Hz. Seismic records can be inverted to determine the location, length and orientation of a lightning channel (Kappus and Vernon, 1991). In the case of the Lauterbrunnen valley, one hour of the seismic record preceding a detected rockfall event was screened for all stations. Lightning was interpreted when the signals showed a sharp blast-like pulse, time offsets between stations corresponding to the speed of sound (about 340 m s$^{-1}$), a wide frequency spectrum peaking between 5 and 15 Hz, and a coincidence with precipitation (assuming there were no dry weather lightning events).

### 3.4.4 Heat-related triggers

Freeze-thaw and thaw-freeze transitions were defined as switches from negative to positive (and vice versa) air temperatures between two consecutive hours. Constraining the thermal effect within rock is not a straightforward task. Direct measurements require intense instrumentation (Collins and Stock, 2016). Geophysical tomography monitoring (Krautblatter et al., 2010) from the rock surface is an alternative but also requires extensive work. Heat diffusion models (e.g., Martinez et al., 2014) can deliver temperature estimates at different levels of spatial and temporal resolution and complexity; however, at first order, air or surface temperature is a valuable proxy for describing the freeze-thaw actions close to the surface of rock masses. Temperature cannot be treated as a regionally constant parameter as air temperature drops by $0.6\,°$ C for every 100 m rise in elevation, and mountain wind systems and topographic shading effects contribute to further modifications of spatial temperature patterns. To relate air temperature as a first order proxy to freeze-thaw action, lag times were calculated for both uncorrected temperature data from the Mürren meteorological station and elevation-corrected values based on the DEM and the seismically located rockfall events.

Constraining thermal stress is similarly demanding as evaluating freeze-thaw action. One needs to link the heat influx (through sunlight exposure and/or ambient temperature) to the thermodynamic properties of the rock medium (e.g., Collins and Stock, 2016). However, excluding the material properties, which mainly control the speed and effectiveness of heat propagation, first order proxies for thermal stress can be provided by the ambient air temperature time series and its first derivative (temperature change rates) as well as spatially resolved sun exposure models, although more complex models are available (e.g., Haberkorn et al., 2016). To investigate thermal stress, the temperature history of rockfall events was described with linear regression slopes of normalised air temperature (Mürren station data) in time windows of 12, 6 and 3 hours before each event occurred. Thermal stress is also linked to the exposure duration of a given section of the cliff to direct sunlight, at the diurnal and seasonal time scale. The Lauterbrunnen Valley, with its almost North-South oriented cliff may exhibit great spatial and temporal variability of exposure time to direct sunlight. The topography-corrected potential exposure time to direct sunlight of the cliff face was modelled with the regional DEM (extending about 30 by 30 km around the instrumented area) and the model insol (Corripio, 2014). Calculations were performed for 1 and 31 March and 2016, yielding the cumulative daily exposure time for each pixel of the DEM and the lowest sunlit elevation along the cliff for a set of hours (8:00, 8:30, 9:00, 10:00, 12:00 am, 1:00, 2:00 pm) through the entire month. March was chosen because it yielded the largest variability in sun exposure of all instrumented months, according to exploratory model runs with lower spatial and temporal resolutions.

### 3.4.5 Biotic/anthropogenic and other triggers

Anthropogenic activity such as construction work, helicopter flights, and rail and road traffic in the valley was observed throughout the deployment and maintenance campaigns and can easily be detected in the seismic records (Dietze et al., 2017). Due to the broad range of possible signals generated by anthropogenic activity it is not straightforward to develop automatic routines to analyse the lag times with rockfalls. Thus, the history of each potential rockfall event was investigated manually up to one hour back in time. Ground vibrations caused by other Earth surface processes were also checked manually by screening

one hour of seismic data before the onset of a rockfall event to identify any signals that could be interpreted as geomorphic activity (cf. Burtin et al., 2014; Turowski et al., 2016, for examples).

All analyses were performed in the R environment for statistical computing (R Development Core Team, 2015) (version 3.3.1) using the packages eseis (Dietze, 2016), sp (Pebesma and Bivand, 2005; Bivand et al., 2013; Pebesma and Bivand, 2016, version 1.2-3), raster (Hijmans, 2016), fields (Nychka et al., 2015), insol (Corripio, 2014) and rgl (Adler and Murdoch, 2016). Dates and times of all events are given with respect to the local time, i.e., UTC minus 2 hours. For the 2015 period, this includes 12 days without daylight saving time (switch was on 29 March), which was ignored here.

## 4 Results

### 4.1 Detection and location of seismic events

During both deployment periods there were always at least four seismic stations in operation, continuously recording data. The STA/LTA picking algorithm yielded initial numbers of 3248 (2014) and 1514 (2015) events. After application of the automatic rejection criteria the number decreased to 603 and 271, respectively. Manual inspection and rejection removed predominantly spurious events (582 and 231), for example related to rail traffic and small earthquakes. The remaining potential rockfall signals (21 and 40) were migrated and yielded a total of 17 rockfalls inside the area of interest for 2014 (ten of them in the period of interest from Dietze (2016)) and 32 for 2015. The remaining rockfalls were located either on the other side of the valley or higher up in the catchment and will not be discussed further. The supplementary material contains a comprehensive table with all detected rockfall events along with their assigned parameters.

The average duration of the picked events was $1.14^{+0.79}_{-0.46}$ s (median and quartiles) in 2014 and $1.56^{+0.67}_{-0.54}$ s in 2015 (global average was $1.37^{+0.84}_{-0.37}$) s, which was clearly different from other signals excluded during the selection process (raindrop impacts and earthquakes). Note however, that the STA/LTA algorithm usually picked the first rockfall impact and all subsequent impacts were rejected from the data set if they occurred within 12 s (cf. section 3). Thus, the STA/LTA-based durations do not represent a realistic estimate of the true event duration ($4.7^{+2.8}_{-2.0}$ s), which has been determined based on manual inspection of waveform and PSD data. An event from 6 April 2015 (figure 2) had a picked duration of 10 s, according to its prolonged activity after the first excursion of the seismic signal. The average SNR of all events was $15.47^{+10.04}_{-2.00}$. For events 41 and 44 (cf. table S1) the per-station-SNR for location had to be adjusted to 6 (exclusion of a spurious superimposed signal at one station) and 4 (inclusion of a low amplitude rockfall signal at one station), respectively.

Earthquake detection for lag time analysis yielded a total of 359 events, lasting on average $14.9^{+28.3}_{-6.4}$ s. The query of the Swiss Seismological Service data base did not yield any earthquake with Mw > 1 within a radius of 20 km during the monitoring period.

## 4.2 Rockfall characteristics

Based on the waveform and PSD data of all events, rockfalls could be categorised into two phenomenological types. Type A events (n = 38, i.e., 78 %) exhibit one or a few short pulses of seismic energy. The pulses last less than 2 s and predominantly exhibit frequencies between 5 and 50 Hz. Type B events (n = 11, i.e., 22 %) have an emergent onset and a longer tail of seismic activity, usually lasting 3 to 6 s, but sometimes up to 20 s. The frequencies are also in the range of 5 to 40 Hz, and sometimes up to 50 Hz. Twenty one events of type A exhibit the subsequent emergence of a type B sequence $2.50^{+2.90}_{-0.90}$ s after the last impact signal. This subsequent phase was best visible at seismic stations along the base of the cliff. Twelve events also showed a subdued signal prior to the first significant pulse of seismic energy, which was mostly visible in waveforms of one or two stations along the cliff top. This signal precedes the first major signal pulse by $2.35^{+1.28}_{-1.00}$ s and lasts for less than a second. Three rockfall events have been selected for a detailed description below because they allow significant insight into the evolution of the processes and illustrate the summary of results given above.

### 4.2.1 Case event I

Records from 6 April 2015, 15:19:00 to 15:25:00 show two distinct seismic events (figure 2). Both events are recorded at all four functioning stations. The first one shows the arrival of two pulses with sharp onsets and a more than one minute long coda. All stations record this pattern in almost the same shape and intensity; amplitudes at station "Funny Rain" are twice as high as at the other stations. The second event is clearly different: the seismogram from "Funny Rain" shows a sharp amplitude excursion for less than 2 s, followed by an emergent onset of activity for 30 s. The other seismograms only show the first pulse. When filtered between 1 and 3 Hz (figure 2 a insets), it becomes clear that the 1.5 second long signal exhibits arrival time offsets between 70 and 450 ms among the stations. Power spectral density estimates of the vertical component of the "Funny Rain" record (figure 2 b) show that the first event is dominated by frequencies between < 1 and up to 60 Hz, with lower frequencies arriving earlier and lasting longer than higher frequencies (triangular pattern). The second event shows a sharp pulse over the entire frequency range above 5 Hz that drops rapidly to a triangular shape of frequencies below 40 Hz. This second signal was a combination of type A and B, with no pause between the two types.

We interpret the recorded events as two fundamentally different seismic sources. The first one shows all characteristics of an earthquake: separated arrival of P and S waves, minimum time offset among the seismic signals, overall marginal signal amplitude differences, a triangular shape of the PSD and a long lasting coda. In contrast, the second event is a common example of a rockfall because the initial short pulse, visible at all stations, covers a wide frequency spectrum, except for frequencies below about 5 Hz. It is followed by signals of subsequent slope activity only at the station at the base of the cliff, which argues for a very local, weak source.

The rockfall is independent of the preceding earthquake, given the more than three minutes long time gap and low intensity of ground movement. The first evidence of the rockfall was a faint signal 1.5 s before the most powerful signal part, which is hard to see in the waveform but clear in the power spectral density estimate (zoomed part of figure 2 b). Whether this faint signal represents the detachment of the rock mass or the release of some smaller rocks prior to the large detachment

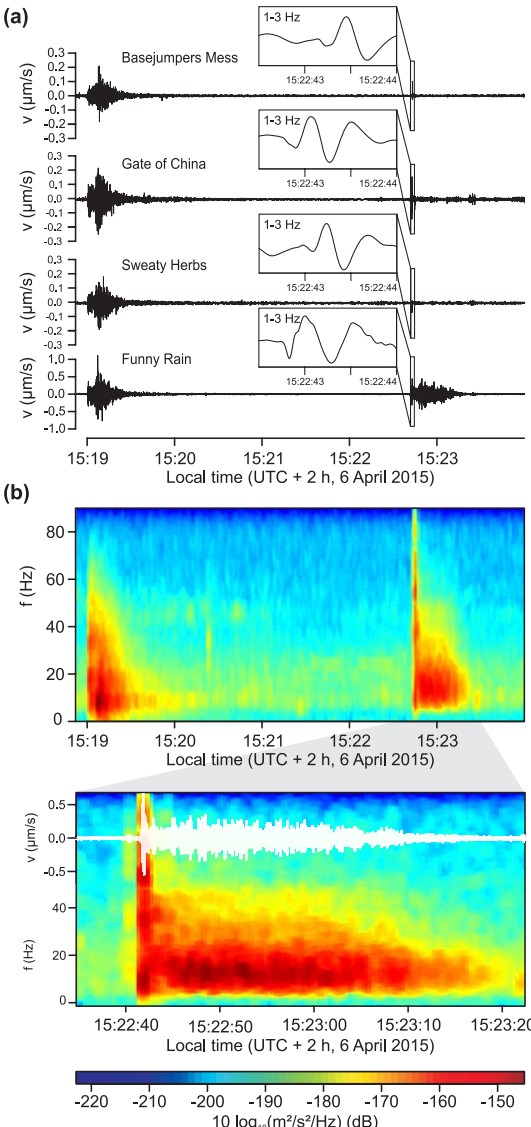

**Figure 2.** Seismic data of two detected seismic events. The first one shows the arrival of two pulses with sharp onsets homogeneously at all stations and a more than one minute long coda. The second one shows very different signal properties across the seismic stations. a: Waveform data (0.5–90 Hz) starting with a small earthquake on 6 April 2015 15:19 that shows the typical P- and S-wave arrivals and coda. 3.6 minutes later, a rockfall (event 30) was detected, showing a very different, distinct waveform pattern. Inset shows lowpass-filtered (1-3 Hz) signals of the initial rock mass impact with clear time offset between the seismic stations. b: Power spectral density estimates of the earthquake and rockfall as recorded by the vertical component of station "Funny Rain". The zoomed part shows the waveform of the rockfall again.

cannot be resolved. The strongest signal, visible at all stations, is interpreted as the actual impact of the rock mass on the cliff face, which leads directly to avalanche-like slope activity for more than 30 s. This last part of the sequence did not give any clear location estimate. The first impact could be located at the northern shoulder of a hanging valley (event 12 in figure 5 c). Apparently, downslope topography was not steep enough to support an immediate free fall phase. A likely scenario is that upon

the first impact the rock mass became fragmented and tumbled down the valley shoulder where it might have become further fragmented, then experienced a free fall phase and started hitting the talus slope as a rain of small rock fragments, lasting for more than 30 s. Thus, the seismic data of this rockfall provide insight to all relevant stages: initiation/detachment, free fall, impact and fragmentation, and continuous slope activity caused by impacting and entrained debris.

### 4.2.2 Case event II

A rockfall of type A with subsequent emergence of type B (figure 3) occurred on 26 October 2014, 22:08:41 and lasted about 45 s in total. Four phases can be distinguished. The first phase was characterised by two short seismic pulses each lasting less than a second. Phase 2 starts 3.52 s after the first one, with a similar double pulse. The pulses are visible at all stations, although the largest amplitudes occur at station "Funny Rain". Predominant frequencies are between 5 and 50 Hz. Phase 3 starts with a sudden onset of seismic activity 3.38 s later and lasts about 5 s. It was also visible at all stations but amplitudes at "Funny Rain"

are 500 times higher than, for example at "Basejumper's Mess". Arising from Phase 3, phase 4 exhibits an almost rhythmic appearance of more than 16 pulses in the waveform of station "Funny Rain" and was not visible at any other seismic station. The pulses are 0.15 to 0.25 s long and separated by pauses of 0.5 to 0.8 s. The amplitudes of the individual pulses rise slowly until they peak at 20:09:03.5 and then fall back into seismic background levels after about 5 s.

Interpreting this case (figure 3) illustrates the potential of environmental seismology to resolve multiple collisions of de-

tached rock masses as well as the high degree of detail to describe the individual process kinetics of single rocks moving through the landscape. The described rockfall, event 8 of the data set of Dietze et al. (2017), is of type A (during phase 1 and 2) followed by a special case of type B (during phase 3 and 4). Phases 1 and 2 are interpreted as two rock mass impacts along the cliff face at subsequently lower positions (figure 3). Phase 3 represents the impact on the talus slope, mobilising a series of rock fragments. Finally (phase 4), one larger rock fragment starts rolling and jumping down the talus slope, towards and

past the station "Funny Rain". The described event has released $0.258 \pm 0.014$ m$^3$ from a spot about 994 m asl., the only detachment area in this section of the cliff visible in the TLS data. The detachment area was just above the seismic station at 888 m asl. and some 20 to 30 m to the North. This implies an TLS-based free fall distance of not more than 106 m (figure 3 d). The seismic estimate of the impact was 919 m asl., with the most likely impact coordinates only 52 m away from the seismic station. Converting the time between the first and second impacts of phase 1 (3.52 s) into fall distance yields 61 m. The time

offset between the second impact and the onset of phase 2 (3.38 s) represents a similar fall distance of 56 m. Thus, in this case, it was possible to show that the two impacts resulted from one rather than two discrete detachment events. The data also gives insight into the mechanism by which impacting rock fragments continue to move downslope (short seismic pulses in figure 3 d).

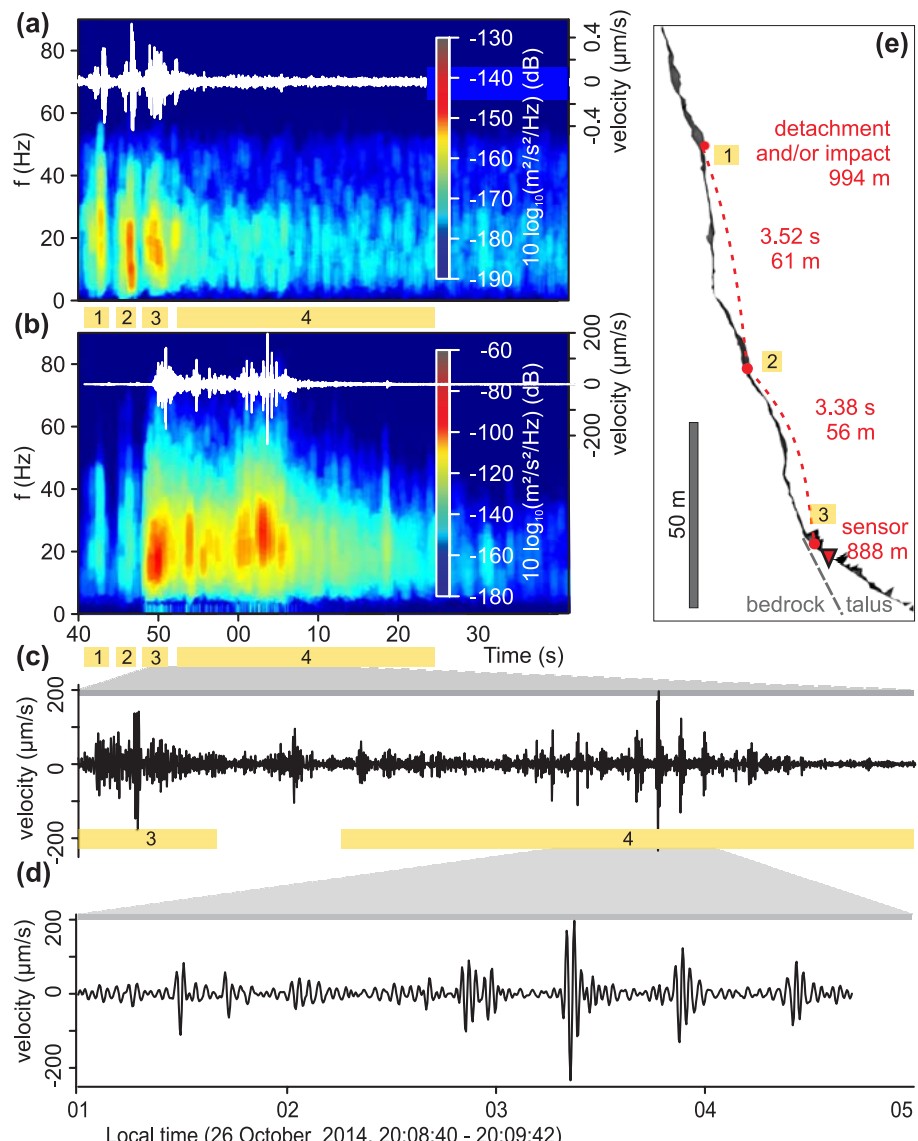

**Figure 3.** Seismic view of a rockfall (i.e., event 17) with multiple impacts and subsequent talus slope activity. Power spectral density estimate and waveform of stations "Basejumber's Mess" (a) and "Funny Rain" (b). The station on top of the cliff (a) mainly records the successive impacts along the cliff face, while the basal station (b) reflects the impact on the talus slope and subsequent slope activity. Note the different range of the colour schemes and waveform ranges in (a) and (b). (c) and (d) show zooms into records of station "Funny Rain". (c) shows the rock mass impact on the talus slope (phase 2) and subsequent activation of a single rock, first rolling then jumping down the slope (phase 3). (d) shows the individual hops of the rock as it approaches and passes the seismic station. (e) shows a profile of the TLS data with detachment area (cf. Dietze et al., 2017), sensor deployment position and free fall times as well as possible trajectories indicated.

### 4.2.3 Case event III

Case event III (figure 4) is a complex sequence of 5 discrete pulses, which can be interpreted as a type A event (pulses 1–3), followed by a type B (pulse 4) and another subsequent type A event (pulse 5). All pulses last for less than 2 s except for pulse 4, which emerges 6–7 s after pulse 3 and lasts more than 30 s until it is no longer discernible from the background noise. Pulse 5 is intersecting with pulse 4. Signal amplitudes for pulses 1, 2, 3 and 5 are high compared to pulse 4 at station "Basejumper's Mess" whereas this pattern is exactly reversed at station "Funny Rain", where pulse 4 dominates. Likewise, with increasing distance to "Basejumpers Mess", all signals decline in amplitude and become more conjoined for the stations on top of the cliff. Only at "Basejumper's Mess" ($\pm$ 0.6 $\mu$m s$^{-1}$) and very faintly at "Gate of China" ($\pm$ 0.03 $\mu$m s$^{-1}$), there is a signal visible that precedes the entire sequence (pulse 0). The same short duration pulses are also visible in the power spectral density estimates (figure 4 b). Pulse 0 ranges between 50 and 80 Hz at station "Basejumper's Mess"; all other pulses cover the full frequency range. In contrast, station "Funny Rain" predominantly exhibits lower frequencies, up to 60 Hz for the short duration pulses and up to 40 Hz for the longer pulse 4, which shows an evolution similar to that of the event described in case 1 (figure 2). Location estimates of the seismic sources were possible for all pulses when adjusting the frequency windows prior to migration of the signal envelopes to the dominant frequency content of the signals (figure 4 c and d) manually based on the spectra of the clipped signals. While estimates for pulses 1–3 overlap, there is a clear distinction to pulses 4 and 5. Pulses 1–3 are located at the central cliff part below the large ledge (1148–1124 m asl.), pulse 4 focuses at the base (922 m asl.) and pulse 5 just below the rim of the ledge (1275 m asl.). The frequency spectra and signal envelopes used for locating the individual impact pulses are contained in the supplementary material.

This last case event (figure 4) illustrates how seismic methods can shed light on the complexity and interaction of processes. The described rockfall is of type A (multiple impacts). Like the first example it exhibits a faint seismic signal (pulse 0 in figure 4), about 1 second prior to the first strong impact of rocks recorded by all stations. By combining the information from signal waveforms and power spectral density estimates, a detailed evolutionary scenario can be interpreted:

- 11:16:24.0 – a weak signal, dominated by high frequency content, was caused by the impact below the large ledge in the central part of the monitored cliff area, at about 1145 m asl.

- 11:16:25.3 – a sharp distinct pulse of seismic energy with highest amplitudes close to station "Basejumper's Mess" was caused by an impact of the failed rock mass at an elevation close to the former location (about 1148 m asl.).

- 11:16:26.3 – another pulse of seismic energy was emitted by the already partly fragmented rock mass hitting the cliff face a further time, just about 20 m below the former spot. Upon this impact, the rock mass was further fragmented and falls freely down the rest of the cliff (calm period of about 6.7 s after the impact signals).

- 11:16:32.5 – a sequence of seismic activity, most intense at the base of the cliff close to station "Funny Rain", was emerging and lasting for 30 s. In contrast to the preceding impacts, there were no erratic pulses of energy but a continuous, asymmetric rise and fall of the entire signal envelope. This was most likely due to first fragments of the initial rock mass

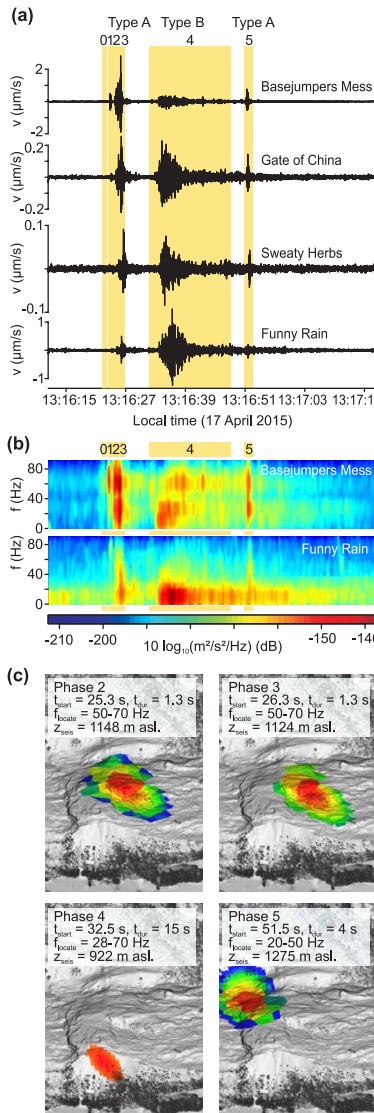

**Figure 4.** Seismic view of a complex rockfall (event 37). a: signal waveforms of the sequence from four seismic stations, filtered between 0.5 and 90 Hz with four distinct seismic pulses and a longer emergent part. b: Power spectral density estimates of station "Basejumper's Mess" and "Funny Rain". The five distinct events cover different frequency ranges over both, time and space. c: Seismic impact location estimates of event 2 to 5 with location polygons clipped at the 0.99 quantile for illustrative reasons. Scenes show an oblique aerial view onto the lidar-based surface model of the affected cliff section. Values denote $t_{start}$ start time (second of the UTC time denoted in a), $t_{dur}$ event duration used as time window for the signal migration approach, $f_{locate}$ frequency range used to filter the data before migration (based on ranges of all stations) and $z_{seis}$ height of location estimate. All times refer to station "Basejumper's Mess".

reaching the talus slope below the detachment area and a subsequent continuous rain of particles, the products of the previous fragmentation of the rock mass, for half a minute.

– 11:16:51.5 – Another short pulse of seismic energy, now above the initial impact zone of the first rockfall, was intersecting with the still ongoing shower of rock fragments. This second sharp signal corresponds to the impact of another rock mass about 125 m above the first one, directly at the rim of the large ledge, some 70 m south of the first detachment area (thus, in total about 143 m away from the first rockfall). This second rockfall might have been triggered by the impacts of the preceding one. Alternatively, and in agreement with the overlapping location uncertainty polygons, this rockfall might have been mobilised from or near to the origination area of the first one.

The location estimates of all events are vague when focusing on the seismic estimates (figure 4 c). In fact, the steep part of the cliff is poorly resolved in vertical direction by only a few DEM pixels, which leads to considerable shifts in the maximum location probabilities. Hence, this case shows the lower limit of location possibilities for such extreme topography. However, when calculating the free fall distances based on the time offsets between the individual impact times the agreement is remarkable. Based on the seismic location estimate, the second rock mass impact was located 24 m below the first one, compared to a distance of 4.9 m calculated based on gravitational acceleration for 1.0 s. Seismic estimates determine the impact at the base, 202 m below the former impact, while gravitational acceleration calculation yields a downslope distance of 220 m after 6.7 s free fall time.

The three examples show the diversity of how rockfalls may evolve and how environmental seismology can provide detailed insights into this geomorphic process, difficult to achieve with other methods in such a holistic way. A posterior mapping approach would have misinterpreted the two potentially linked events from case 3 (figure 4) as two discrete rockfalls. Likewise, the seismic approach could resolve multiple releases of rock masses from the same detachment area at different times, which would be amalgamated to one larger rockfall event by other methods. Video imagery would have allowed event detection, location and possibly also insights to event anatomy, given that weather (clouds/fog, rain, snow cover) and daylight conditions were suitable, rocks did not move into tree covered areas, and the events were large enough to be resolved, which in turn limits to area that can be surveyed.

## 4.3 Spatial activity patterns

The 17 rockfalls recorded in 2014 mostly occurred in the lower southern part of the instrumented cliff section (figure 5 a and c). A minor centre of activity was in the northern part of the cliff. In contrast, most of the 32 rockfalls detected in 2015 occurred at the upper and central parts of the cliff, in the central and northern section (figure 5 b and c). There appear to be three activity hotspots in 2015. The southern one was located just north of the hanging valley of the Ägertenbach (figure 1) and comprises four events. The other two are below the edges of the large ledge in the central part of the steepest and longest cliff section between stations "Gate of China" and "Basejumper's Mess". There, the southern one comprises ten, the northern one six recorded events. However, the other parts of the instrumented area were also affected by single rockfall events (dark blue coloured patches in figure 5 a and b). Five impacts occurred at the base and four near the top of the cliff (figure 5 c).

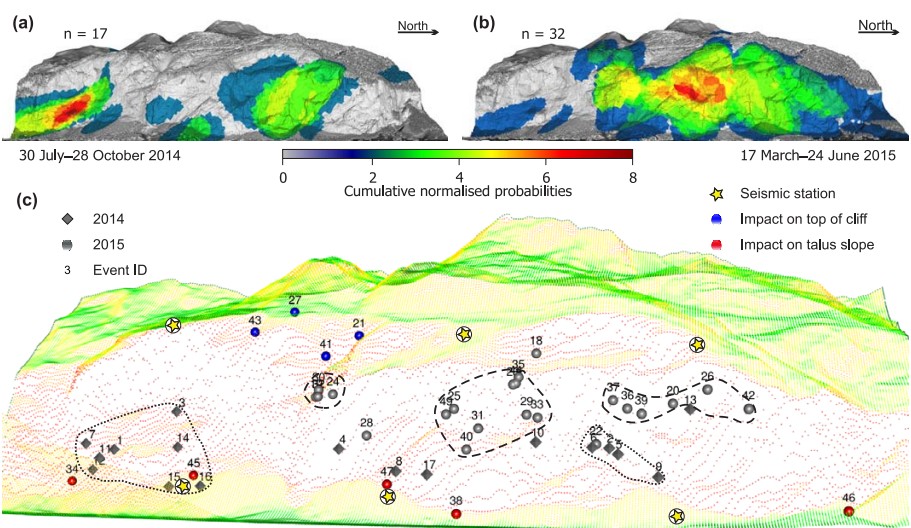

**Figure 5.** Spatial rockfall activity patterns. a: 3D scene view of compound seismic location estimates for summer/autumn 2014. b: 3D scene view of compound seismic location estimates for spring 2015. Normalised probabilities based on summation of normalised location probabilities > 97 % for all detected events. c: 3D point cloud view of DEM pixels coloured by normalised slope inclination (note the sparse coverage of the steep cliff parts) and maximum location probability of rockfall events as numbered spheres. Blue spheres indicate events at or directly below the cliff edge, red spheres denote events with impact locations at the talus slopes and black spheres denote events in the central cliff part. Cube symbols depict 2014 events, sphere symbols denote 2015 events. Dotted lines encircle 2014 activity hotspots, dashed lines show 2015 hotspots.

## 4.4 Temporal activity patterns

Rockfall activity was distributed over almost the entire instrumented period in both years (figure 6 a and b). However, the distribution was not uniform. In 2014, ten of 17 detected rockfalls occurred in a four-week window during 12 weeks of recording, and three weeks from 2 to 25 September had no activity. Mostly, activity occurred in clusters of two to three events.

5  These patterns of rockfall timing were similar in 2015 although activity was greater and concentrated in the first five weeks of the monitoring window. There were three periods of enhanced activity that account for two thirds of all events: 19–21 March (7 events), 6–9 April (6 events) and 17–21 April (7 events). The rockfalls in these three periods were not clustered in space. There were always more than three cliff sections affected per period and location estimates were never horizontally closer than 40 m to each other, i.e., events close in time were separated by several hundred metres.

10  When normalised by cliff area (2.16 km$^2$) the average event rate over the entire instrumented period in 2014 was 2.64 rockfalls per month per km$^2$. On a month by month basis, we obtain values of 2.78 (August), 0.93 (September) and 4.17 (October). In 2015 the average rate was 5.01 rockfalls per month per km$^2$. For the four individual months rates were 11.50 (March, only 15 days included), 6.96 (April), 2.32 (May) and 4.11 (June, only 7 days included) per month and km$^2$.

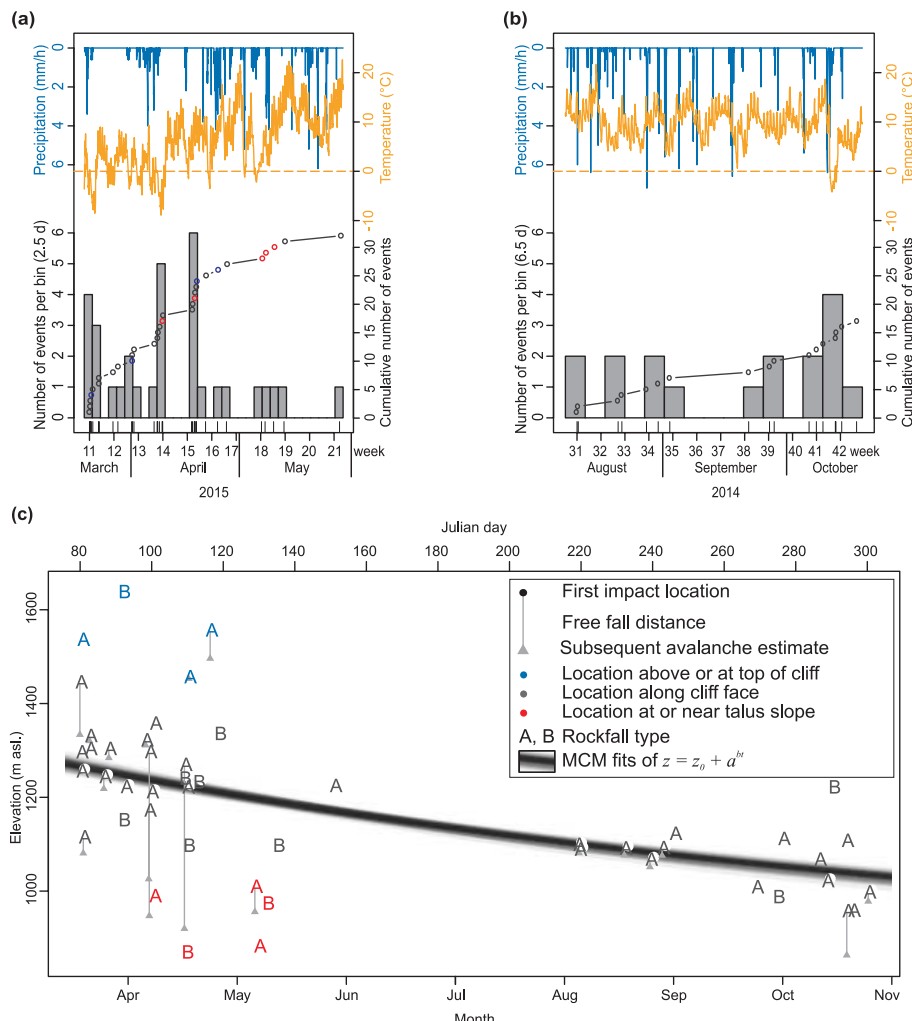

**Figure 6.** Temporal patterns of rockfall activity. a: times series of all detected rockfalls from spring 2015. b: time series of summer/autumn 2014. Histogram bars and rug show rockfall events, circle-line graph shows cumulative number of events. For the colour of circles, see legend to c. Dashed orange line depicts zero degree. c: Height of the seismically estimated first rock mass impact as function of month of the year (note that 2014 events are plotted right of the 2015 events). Grey vertical lines and triangles give projected downslope displacement (where possible) due to gravitational acceleration and time offset to subsequent avalanche emergences. Letter denotes phenomenological rockfall type. Semi-transparent graphs are Monte-Carlo based exponential fits of the elevations of the events denoted in grey colour.

## 4.5 Lag time analysis

For all relevant trigger mechanisms the lag times of the 49 detected rockfalls were individually analysed using kernel density estimates (figure 7). These estimates stretch differently in time, depending on the maximum lag times identified in the data.

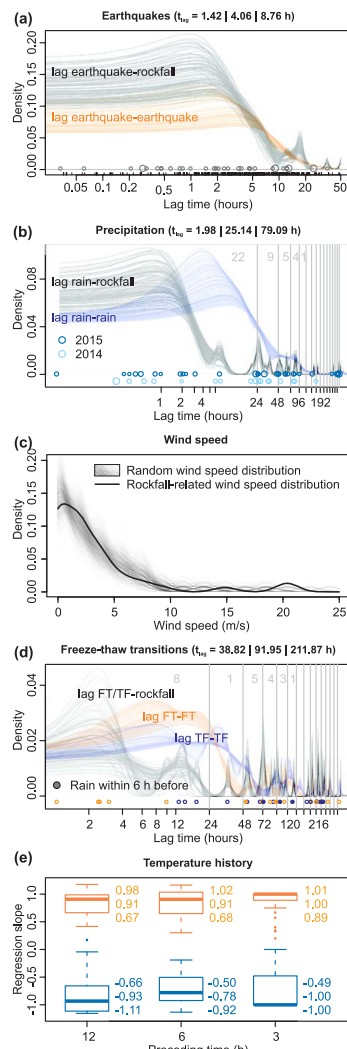

**Figure 7.** Time lags of all detected rockfall events to different potential triggers as shown by kernel density estimates based on subsampling the rockfall data set and using different kernel sizes. Values in brackets in titles denote 25, 50 and 75 percentiles of the lag times. a: Earthquakes. Lag times are of the order of lag times between earthquakes (thick black line). Circle size is proportional to seismic energy of the earthquake signal envelopes. b: Precipitation. Circle size proportional to cumulative rain amount of a preceding event. Circle colour depicts year. c: Wind speed. Thick black line depicts wind speed distribution during rockfall events, grey lines show distributions of randomly selected wind speed samples. d: Freeze-thaw events. Circle colour indicates freeze-thaw (orange) versus thaw-freeze (blue) transitions. Filled circles indicate combination with rainfall within six hours. e: Temperature history before an event, illustrated as slope coefficients of linear regression lines of normalised air temperatures 3, 6 and 12 hours before a rockfall event. Positive (orange) and negative (blue) slopes are separated by colour. Values next to box plots denote quartiles.

Usually they are polymodal but the first mode, always within 24 hours, is the dominant one in all curves. Note that the time axes in figure 7 a, b and d are in logarithmic scale to focus on this first mode in the density curves, most important for rockfalls, rather than showing the properties of the full distributions.

The density estimates of earthquake lag times, i.e., the time passed between an earthquake and the next occurring rockfall, (figure 7 a) peak between 1 and 2 hours, which is almost equal to the lag time between two earthquakes. One rockfall occurred 2 minutes after an earthquake, a second one 3.7 minutes later (i.e., the event from figure 2) and all others at least 12.5 minutes after an earthquake. All these earthquakes were very small local events. The strongest earthquake-related peak ground acceleration value measured by the seismic array was $0.05$ ms$^{-2}$ (i.e., $5.1 \cdot 10^{-3}$ g) and was more than 8 hours before the next rockfall event. The strongest earthquake from the data base of the Swiss seismological survey within the queried radius of 20 km from the centre point of the cliff was near Brienz (29 March 23:43:22) with Mw = 0.9, occurring about 65 hours before the next rockfall.

The lag time density estimates for precipitation events above $0.1$ mm h$^{-1}$ peak at 1 hour. Out of the 49 events, 11 (i.e., 22.4 %) occurred within one hour, and 22 (i.e., 44.9 %) within one day after a rainfall event. There was no difference between the 2014 and the 2015 data and also no systematic trend between lag time and cumulative rainfall amount ($R^2 = 0.09$, cf. circle size in figure 7 b).

Wind speed during rockfall events ranged from 0 (17 events) to 20.4 (2 events) ms$^{-1}$. The distribution function of wind speed during rockfalls does not differ from the 1001 distribution functions of each 49 randomly selected hours throughout the data set (figure 7 c).

Lag times for freeze-thaw related rockfalls (figure 7 d) peaked between two and three hours. Thaw-freeze-related events showed lag times of around 12 hours, i.e., they occurred about half a diurnal cycle before a rockfall occurs. Likewise, there were constant, strong linear temperature trends 3, 6 and 12 hours before a rockfall event (figure 7 e). The slope coefficients of the trend lines were closer to one for the rising temperature trends than for the cooling trends. Hence, temperatures rose or fell nearly linearly for 12 hours before a rockfall occurred.

Manual screening of the seismic records revealed no signals of lightning strikes within the period of one hour before a rockfall event (cf. supplementary materials for the full list of manual screening results).

# 5 Discussion

## 5.1 Rockfall characteristics

There are two distinct types of rockfall signals. Type A is interpreted as rocks that are released and experience a free fall phase before colliding once or multiple times with the cliff. Except for event 16 (cf. table S1 or figure 5), events of this type did not hit the base of the cliff or the talus slopes directly (figure 6 c). Indeed, half of the type A events exhibit the emergence of a prolonged signal some time after the initial impact, which may be best explained by rock fragments that subsequently reach the cliff base. The event in figure 2 is an example of rockfall type A with a very short pause between first impact and subsequent emergence of the rock fragment avalanche. Multiple impacts can be either caused by the same initial rock volume at subsequently lower parts of the cliff (i.e., example from figure 4) or by different rock masses subsequently detaching from

the same source region. For most of the cases it was not possible to distinguish between these two possibilities. Events of type B are interpreted as avalanches of rocks (e.g., prolonged activity in figure 2 b, individual pulse 4 from figure 4, phase 3 from figure 3). Events that are entirely of this type are generally lacking a distinct initial impact of a free falling rock mass. They occur at the base (event 38 and 47), on top of the cliff (event 41), just above the large ledge (26) or near other, less steep or

step-structured parts of the cliff (event 39, figure 6 c). In summary, the events from the Lauterbrunnen Valley exhibit a range of rockfall scenarios, depending not only on the detachment height but also on the geometry of the cliff, i.e., whether surface topography supports free fall of rocks or causes avalanche-like translocation. Likewise, rockfall activity in the Lauterbrunnen valley does not necessarily trigger talus slope activity, the subsequent transport corridor of the sediment cascade. Rather, rock masses with such small volumes apparently can be accumulated on the talus slope without destabilising it immediately.

The time difference between the first clear rockfall related signal and the emergence of prolonged avalanche-like activity can be interpreted as free fall time. Converting time to vertical displacement due to gravitational acceleration yields fall distances between about 1 and 286 m, excluding obvious outliers (events 8, 32, 34, 35 and 37 with time differences between 8.4 and 21.0 s) that probably represent remobilisation of material on talus slopes after some pause.

    Whether seismic signals preceding the first clear rockfall impact signal are related to the detachment process (Hibert et al.,

2011) or are a result of a first, low energy impact of an already detached rock mass cannot be resolved here. For rock volumes predominantly below 1 m$^3$ the elastic rebound of the cliff is weak and more energetic signals require some free fall of the rock mass before an impact. The general cliff geometry (88.5 °inclination and structured by small ledges) certainly supports low energy impacts just after detachment. Nevertheless, location of the first clear impact signals will always pick a spot below the actual detachment site. Thus, when converting the time delays between the twelve observed rockfall initiation signals, may it

be detachment or low energy impact, and the first clear impact signal ($2.35^{+1.28}_{-1.00}$) to fall distance we would need to add $27^{+37}_{-18}$ m to each event to get a more realistic estimate of the detachment height. This value is well within the location uncertainty range. Thus, we do not correct the location estimate for this effect.

## 5.2   Spatial and temporal activity patterns

Rockfall impact areas from seismic monitoring are scattered across the entire area of observation but show distinct horizontal

(southern part in 2014 versus three central clusters in 2015) and vertical (basal parts in 2014 versus central and upper parts in 2015) patterns. This short-term variability highlights the necessity to resolve sub-annual time-scales of activity, even below seasonal survey recurrence intervals. 59 % of all events occur in 12 % of the instrumented time in 2015, during three discrete activity periods. Within each of the activity periods the impacts are predominantly laterally spread by several hundred metres.

    Except for one event that stretches into the upper part of the Ägertenbach waterfall, localities of the 2015 data are outside

of the hanging valleys where collapsing frozen waterfalls may act as source of seismic signals that might be misinterpreted as rockfalls. The seismic array also detected rockfall events outside the monitored cliff face. These were mainly from two other active areas: the west-facing valley side and the steep east-facing slope of the Chänelegg above the town of Mürren (figure 1 a).

    When comparing the seismic-based spatial activity patterns from this survey with the laser scan-based patterns of Strunden et al. (2014) some similarities can be found despite the rather long integration times of the laser scan campaigns. During

June–December 2012, 16 rockfalls were released, mainly at the lower cliff part. In contrast, during January–March 2013 the 19 detected rockfalls came from the central and upper part of the cliff. Interestingly, there were no events in the areas of the three activity hot spots of the 2015 seismic monitoring period.

Exploiting the seismic data with much better temporal resolution combined with the location estimates, the TLS-based pattern becomes a clear trend. The rank correlation coefficient between Julian day and rockfall activity elevation is $\tau = -0.56$ ($p = 5 \cdot 10^{-7}$). However, a linear trend is not the most appropriate model to describe the data since rockfall activity must stop at the base of the cliff. Thus we fitted an exponential model of the form $z = z_0 + a^{bt}$ with $z_0$ being the lowest elevation of rockfall activity, i.e., the valley floor at an average elevation of 850 m asl., $t$ being the day of the year, and $a$ and $b$ being parameters to estimate. To better visualise the uncertainty inherent to the modelled data set (figure 6 c) we performed Monte Carlo-based fits of 1001 subsets of the events, each containing 80 to 100 % of the full data set. The model parameters are $a = 566 \pm 45.3$ and $b = -3.6 \cdot 10^{-3} \pm 5.4 \cdot 10^{-4}$. Thus, the exponential term grades from greater than 0.75 at the beginning to about 0.28 at the end of the year. The model predicts an average rockfall activity elevation range of 1277 to 1043 m asl. with a root mean square error of 76 m. Thus, there is significant scatter in this overall trend, underlining that the model only describes a first order effect visible in the data, which is modulated by further factors of influence that impose a strong stochastic effect. It remains unclear when (which time of the year) and where (upper limit of activity) the cycle of seasonally lowering rockfall activity exactly starts without a considerably longer instrumentation period. To shed light onto potential forcing mechanisms of this pattern, we need to first identify the role of trigger mechanisms.

## 5.3 Trigger mechanisms

Several time scales need to be considered when addressing the relationship between rockfall activity and potential triggers. There may be longer scales (e.g., cyclic adaptations of climate at the order of years to millennia) but the largest one visible in this study – though not completely resolved – is the seasonal scale. In this scope, seasonal scale is a scale that focuses on the evolution of patterns over several months. It should not be mixed with the term seasonality, which would focus on the properties and dynamics of such patterns over a period of many repeated seasonal cycles. The seasonal scale sets the constraints for the effectiveness of individual triggers. For example, freeze-thaw transitions may be expected during winter and spring rather than during summer. Superimposed there is a scale at the order of several days to a few weeks, which mainly reflects the actual weather conditions. Further, there is a diurnal scale that alters weather-dictated effects, mainly through the consequences of sunlight exposure. Finally, there is another small-scale modification of activity patterns, related to the response time of the rock mass to the trigger conditions. This scale is of the order of a few seconds to several hours (cf. section 2). Apart from these nested temporal scales in which rockfall triggers manifest, there are also triggers that are completely independent, such as earthquakes, propagation of cracks and anthropogenic activity.

### 5.3.1 The seasonal scale

The seasonal scale is resolved in this study only with two distinct time periods, late summer to autumn (grading from the highest towards moderate temperatures and from the moistest to the driest conditions) and late winter to spring (grading from

the lowest to moderate temperatures and from frozen to liquid water dynamics). The last four rockfalls in 2014 occurred after a temperature excursion below 0 ° C. The first two periods of enhanced rockfall activity in 2015 were associated with freeze-thaw cycles and rainfall. Accordingly, freeze-thaw-related rockfalls occur only in the late autumn and early spring period. The time lags for both free-thaw and thaw-freeze transitions are in agreement with the < 20 h rockfall response times identified by

D'Amato et al. (2016). However, four out of the five rockfalls with a freeze-thaw-transition time lag below about half a day have precipitation lag times of less than one hour, which makes it difficult to argue for temperature as the predominant trigger at this seasonal scale. Thus, even though our approach allows event and trigger timing at hourly resolution it is not possible in these cases to separate the two triggers.

### 5.3.2    The weather event scale

The meteorologically dominated scale is expressed by the three activity periods in 2015 that coincide with strong shifts in temperature (sometimes below zero degrees) but mostly with precipitation events (fig. 6). Accordingly, the precipitation-related lag times, peaking around 1–2 h (fig. 7 b), suggest a strong link between rain and rockfall occurrence. Other studies found similar strong links between rockfall (Helmstetter and Garambois, 2010; Hibert et al., 2014; D'Amato et al., 2016) and other mass wasting process (e.g., Burtin et al., 2013) activity and precipitation. However, the small lag time implies that a temporal

resolution of several hours (D'Amato et al., 2016) is still insufficient to constrain precipitation as trigger. Perhaps even the hourly aggregated meteorological data used in this study is not detailed enough.

The meteorological time series contains 108 rainfall events with at least 0.1 mm/h cumulative precipitation (0.2 was the minimum cumulative amount recorded preceding a rockfall in this study). However, this does not mean that every second rainfall event caused a rockfall. In September 2014 and June/July 2015 there were multiple rainfall events without any rockfall.

Vice versa, the two prominent rockfall episodes, in late May and late April, were not associated with any rainfall or with an exceptionally strong rainfall event (cf. figure 6).

Wind speeds during rockfall events do not differ from random distributions. The overall calm conditions (35 % of the events occurred during zero wind speed, speed among all rockfall events is about 4 m s$^{-1}$) do not render wind a plausible trigger for rockfalls in this study area.

### 5.3.3    The diurnal scale

Nested into this meteorological framework, bulk rockfall activity shows a somewhat bimodal diurnal pattern, peaking at 8 am and 8 pm (figure 8 a, grey lines). Arguably, the density estimates of the event distribution come close to sampling a random event distribution in time (i.e., a Poisson process). Thus, one explanation of the diurnal pattern of rockfall occurrence is that it is a completely random process. However, the time lag analysis from above as well as the discussion of triggers in section 2

already point at a series of underlying mechanisms that influence the likelyhood of a rockfall to occur.

Accordingly, when grouping the events based on their lag times to meteorological phenomena, the bulk pattern of the density estimate curve changes (fig. 8 d). The 16 strongly precipitation-related rockfalls (i.e., events with lag times smaller than 4 h, coinciding with the significant drop of the KDE, cf. fig. 7) form a bimodal distribution with modes at 3–8 am and 6–10 pm,

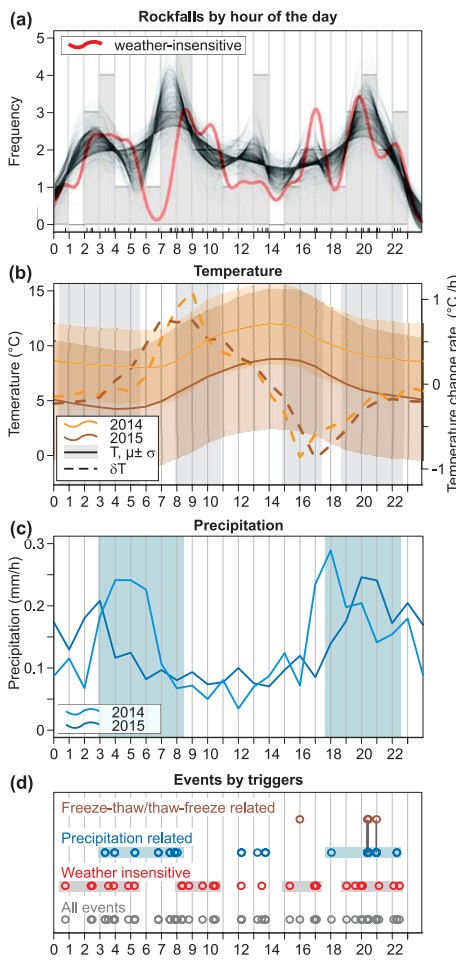

**Figure 8.** Rockfall activity and potential drivers/triggers grouped by the hour of the day. a: Histogram with 1 h wide bins is overlaid with Monte Carlo-based kernel density estimates (grey lines, kernel size changed between 0.5 and 2 h) and a deterministic KDE (kernel size 0.8 h) of the weather-insensitive events. b: Daily air temperatures (solid lines and polygons) and temperature change rates (dashed lines) for summer/autumn 2014 and spring 2015. c: Average precipitation for the two instrumented periods. d: Individual events grouped by drivers. Freeze-thaw-related is defined as time lags < 4 hours (cf. density drop after that time in fig. 7 d). Precipitation-related is defined as time lags < 4 hours (cf. density drop after that time in fig. 7 b). Weather-insensitive is defined as being neither freeze-thaw- nor precipitation-related.

which evidently reflects the overall rainfall pattern at diurnal scales regardless of the season. Subtracting these precipitation-related and the freeze-thaw-related events from the global data set yields only those events that are not related to weather phenomena, keeping in mind that wind speed during rockfalls is not different from random wind speed and that there were no signals of lightning strikes visible in the seismic data. These 26 weather-insensitive rockfalls can be tentatively assigned to
5  four groups (grey polygons in fig. 8 d) that do in turn correspond to the diurnal temperature cycle (fig. 8 b). Namely, the group

from 0–6 am corresponds to the coldest hours of the day, just before daylight, when thermal contraction of the rock is highest and causes the highest stresses. The group from 8–11 am lags the strongest positive temperature change rates by 1–3 h. This correspondence reflects the strongest stress increase due to thermal input. In analogy, the group from 3–5 pm represents the opposite to the former case, with negative temperature change rate. The last group from 7–11 pm appears to be independent

from thermal forcing. Arguably, the number of observations designated to be weather independent is too small to support statistically testing whether the combined diurnal forcing (temperature and temperature change rate, adding to an almost flat probability density distribution with four modes in 24 hours) is a proper model. However, from a mechanistic point of view it would be misleading to assume rockfalls are randomly distributed across the day when they show obvious lag time bounds to environmental conditions or first order physics can explain the stress patterns (Collins and Stock, 2016). Thus, we consider

the above interpretation as one out of perhaps further solutions, though a plausible on based one the two first order effects of diurnal thermal forcing.

Hence, from this detailed insight into the relations of individual events to potential meteorological and solar drivers there appear to be three relevant and independent causes of rockfall in the Lauterbrunnen Valley: i) insolation and heat diffusion that drive thermal expansion and contraction of the rock (17 of 49 rockfalls), ii) precipitation (19 of 49 rockfalls), and iii) freeze-

thaw transitions, perhaps combined with precipitation (5 of 49 rockfalls), leaving 8 rockfalls triggered by other mechanisms or with longer lag times to the above triggers.

### 5.3.4 Time scale-independent triggers

Earthquakes appear to be irrelevant for rockfall activity in the Lauterbrunnen Valley. Although the lag time of a rockfall to an earthquake is between one and two hours and can be as short as a few minutes, this relationship is spurious and reflects

the recurrence time distribution of earthquakes rather than the link to rockfalls (figure 7 a). The strongest recorded nearby earthquake (Mw 0.9, $< 4 \cdot 10^{-3} ms^{-2}$ at "Gate of China") is hardly able to cause any major ground motion in the study area (Meunier et al., 2007; Jibson, 2011), as is also reflected by almost three days until a rockfall occurred. Interestingly, 9 of the 49 events showed the seismic signature of a helicopter passing by, 10 to 5 minutes before a rockfall occurred. However, helocopters cause only small ground accelerations of $< 10^{-3}$ ms$^{-2}$, making a direct influence unlikely. Also the other identified

anthropogenic signals prior to rockfall activity (cf. supplementary materials), such as train signals, blasts and further signals that cannot be clearly assigned to a clear process, always happened several minutes before a rockfall.

### 5.4 Cause of the vertical rockfall activity trend

In spring, when the freeze-thaw trigger is relevant, only the upper parts of the cliff are active, as these receive sufficient sunlight to drive transitions between ice and liquid water. Indeed, through the course of the month of March the upper rim of

the Lauterbrunnen Valley can potentially receive from 8 to 12 hours of sunlight per day, while the cliff base receives only from as little as 3 to about 8–10 hours of sunlight. Most parts of the cliff receive less than 4 hours of sunlight at the beginning of March and can gain between 6 and 12 hours at the end of this month. The lower limit of the sunlit part of the cliff continuously lowers throughout March, especially considering the early hours of a day (figure 9 b). For example, at 8 am there will never be

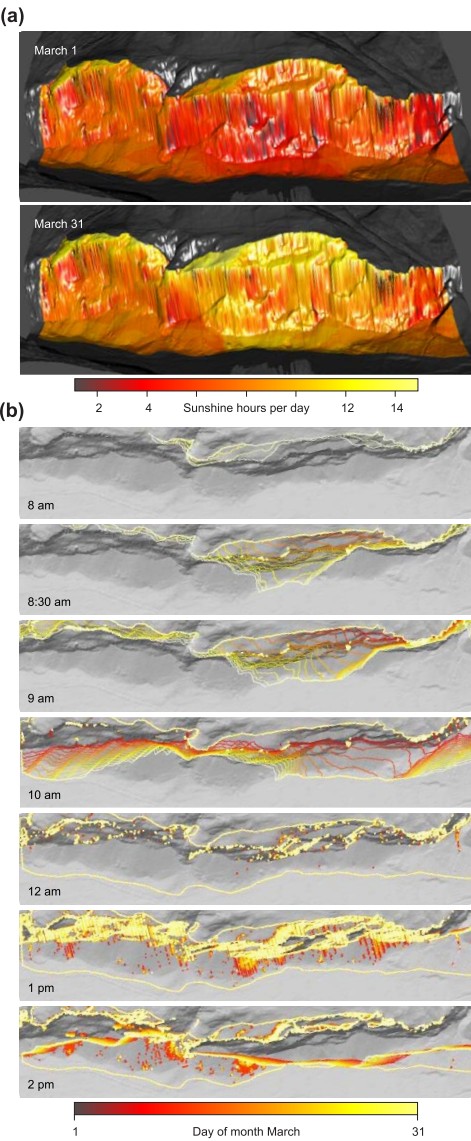

**Figure 9.** Potential sunshine coverage of the western Lauterbrunnen cliff face. a: Cumulative potential sunshine hours for the beginning and end of March 2015. The central part of the cliff shows the most changes, from 2–6 hours on 1 March to 6–14 hours at the end of this month. b: Sunlight-covered areas along the cliff face for different hours of the day (individual panels) and through the course of March 2015 (colour of the lines). The cliff cannot receive sunlight before 8 am, is completely in the sun by noon and in the shadow again around 1 pm. Again, the central cliff part is the most sensitive between 8 and 10 am. Sunlight polygons clipped to area of interest from figure 1.

sun on any part of the cliff regardless of the day of the month, whereas at 8:30 am at the beginning of March only the top part of the cliff is exposed to sunlight and by the end of the month the entire cliff is exposed. Thus, throughout early spring, middle

and lower sections of the cliff are consistently longer exposed to sunlight and the corresponding heat input. During later parts of the year, as the sun angle increases further, this disproportional pattern diminishes. Accordingly, the most likely time for resetting the downward activity shift to the upper part of the cliff should be late winter to early spring, as has been proposed for other rockfall-prone alpine environments (e.g., Matsuoka and Sakai, 1999). During that time only the upper parts of the cliff
experience numerous freeze-thaw transitions (cf. 6 b) and thereby loose the ice as cohesive crack filling.

    For the rest of the year, where we also see the downward trend of rockfall activity with time and differential sunlight exposure can no longer be responsible, another mechanism is required. More specifically, this mechanism must include the sensitivity of the cliff to precipitation events and thermal stress due to heat input and diffusion. We see the most plausible underlying mechanism in a continuously lowering drying front along the cliff face, which is restored during late autumn to early spring
when the cliff is less continuously exposed to sunlight, the major agent of external drying of the rock wall. Water storage is also refreshed by snow melting higher up in the catchment, which provides a more or less continuous supply of water that can seep into the karstic limestone plateau on top of the cliff during the melt season. Field observations are consistent with this drying hypothesis, as seepage out of the cliff is widespread in March, but by August/September, the cliff is dry outside of precipitation events (cf. figure S1). The presence of a vertically shifting window rather than a continuously widening band of
activity suggests a limited potential of a cliff area to release rockfalls once it is appropriately stimulated by a trigger mechanism. In other words, once a given section of the cliff is devoid of all loose rock mass it needs considerable time (at least until the next lapse of the annual cycle) to allow block production processes, such as weathering, dissolution or crack propagation, to create new mobile material that can be released by a trigger mechanism. Thus, while the dry front moves downward it continuously exposes new cliff sections to the action of trigger mechanisms that are able to cause rockfalls sufficiently fast to keep track
with a downward shift of 33 m per month.

    But what is the link between a transition from continuously wet to predominantly dry internal rock state and rockfall susceptibility to precipitation and thermal stress? Precipitation leads to a saturation of the rock mass from the surface inwards, provided the rain event is sufficiently long and intense. However, infiltration and migration of the wetting front into the rock mass only occurs if the medium is not yet saturated, i.e., has a negative matrix potential. Thus, only already internally dry cliff
sections can experience cyclic wetting and drying, a pattern we assume to support destabilisation of rock masses and ultimately rock detachment. Thermal stress is a function of temperature change, which in turn depends on heat input, heat conductivity and heat capacity of the medium. The latter parameter takes a value below 1000 J kg$^{-1}$ K$^{-1}$ for limestone, but more than 4000 J kg$^{-1}$ K$^{-1}$ for liquid water. Thus, as soon as the limestone cliffs contain water, their heat capacity increases, and consequently, their susceptibility to thermal stress drops significantly. This trend gets even stronger when assuming water circulation, which
leads to effective conveyance and extensive dissipation of heat. Thus, a dry limestone cliff section experiences significantly higher temperature amplitudes and, accordingly, thermal stress inside the rock mass.

    Apparently, there is an overlap of the freeze-thaw driven rockfall activity and precipitation-controlled events, whereby the former system is only relevant in the spring period. This pattern is clearly reflected in the monthly aggregated rockfall rates, which range from 11.50 events per km$^2$ in March and 6.96 events per km$^2$ in April to values between 4.17 and 0.96 events per
km$^2$ for the other instrumented months.

# 6    Conclusions

The ability of a seismic network to provide spatial and temporal information on catchment-wide rockfall patterns as well as profound insight into individual stages of single events renders seismic monitoring a universal tool to investigate an important geomorphic process that is otherwise hard to constrain. Insight into event anatomies sheds light onto the variability of the rockfall process. It is possible in several cases to detect rockfall activity even before the first network-wide registered impact signals. The number and consequences (e.g., fragmentation) of individual impacts can in principle be resolved. Time spans between these impacts allow calculation of free fall distances and provide the base for exploring energetic relationships of this mass wasting process. The modes of subsequent slope activity – instantaneous deposition, debris avalanches, single rock jumps – allow assessment of process coupling and connectivity analyses. Combining this detailed information about each event reveals that rockfall in the Lauterbrunnen Valley can cause one or more discrete contacts of a detached rock mass with the cliff or a rather avalanche-like movement of multiple rock fragments.

Rockfall detection and location allows insight to the temporal and spatial variability of rockfall events well below sub-annual time scales. Although this study only provides a first glance at the spatial and temporal variability of rockfalls in steep alpine terrain, and much longer deployment periods are needed (e.g., to cover at least two full annual cycles), the patterns emerging from the highly variable nature of events could give essential input to rockfall susceptibility models and help improve early warning or mitigation strategies. During different seasons rockfall affects laterally and vertically distinct sections of the cliff. More specifically, spatially different sunlight exposure patterns are only relevant to understand freeze-thaw-related rockfalls during a small time window in spring, whereas the most likely cause for a continuously downward shifting window of rockfall activity over the year seems to be a lowering water table inside the limestone cliff. This implies a spatial and temporal interplay between block production (i.e., transformation of stable cliff sections to rockfall prone entities by ice segregation, thermally driven crack expansion or limestone dissolution) and water and heat related activation of the prepared sections by episodic and diurnally forced trigger mechanisms.

We quantify the relative effectiveness of rockfall triggers; based on the high temporal resolution the overlap effect of different triggers can basically deciphered, except for the three cases where freeze-thaw and precipitation lag times are too close. Accordingly, freeze-thaw transitions account only for 5 (10 %) rockfalls, though precipitation perhaps also plays a role for these, and this trigger is only important during a few cold months of the year. Precipitation is relevant for 19 (39 %) rockfalls year round and 17 (35 %) rockfalls are triggered by diurnal temperature changes although through different mechanisms: 7 (41 %) of these 17 events occur during the coldest hours of the day due to contraction of the rock mass and the highest tensions along crack boundaries, 6 (35 %) occur when the heating rate is highest, i.e., when thermal expansion stress rate is highest, and 4 (24 %) occur during the highest cooling rates, i.e., the opposite direction of the former process. Beyond these 17 rockfalls another 7 events occured during the first half of the night without any identified cause. When focusing on the precipitation-related lag time, 11 (22 %) of all rockfalls occur within 1 hour after precipitation and 22 (44 %) within 24 hours. In other words, almost half of the rockfalls can be reduced by avoiding hiking or other activities in rockfall-prone areas for a day after a precipitation event.

For all lag time studies to investigate trigger roles, the final temporal resolution is given by the lowest resolution of all applied techniques: rockfall detection technique, meteorological data, seismic event catalogue, and information about human activity. So far, hourly resolution is the best that could be achieved, limited by the time resolution of the available meteorological data. One way to go beyond this, if there is no chance to increase the temporal resolution of this auxiliary data set, is by substituting the meteorological data by seismic signals of precipitation (Roth et al., 2016) and the temperature data automatically registered at the Cube[3] data loggers. Both parameters can be measured at arbitrary high temporal intervals and, more importantly, at each seismic station. This would allow for much better spatial resolution of the meteorological boundary conditions.

## 7   Data and code availability

The supplementary material contains the raw seismic traces from the recording seismic stations of all rockfalls with a time buffer of 30 s before and after the detected events. The raw point cloud data from the TLS survey is available upon request.

*Author contributions.*  Michael Dietze contributed to seismic fieldwork and data analysis. Kristen L. Cook contributed to terrestrial laser scanning and seismic station maintenance. Jens M. Turowski and Niels Hovius contributed to equipment provision, field work planning and data analysis. All authors contributed to manuscript preparation.

*Acknowledgements.*  The field work campaigns generously benefited from the support of Maggi Fuchs, Michael Krautblatter, Torsten Queißer and Fritz Haubold. The authors are thankful for these creative involvements. We are also thankful to the GIPP seismic device pool for providing six TC120s sensors and Cube[3] data loggers. Solmaz Mohadjer and Todd Ehlers are thanked for joined field work and data discussions. Christoph Burow is thanked for providing the missing puzzle piece. We further thank Angès Hemstetter, Didier Hantz and an anonymous referee for their input and comments.

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
