# Peer review of "Spatiotemporal patterns, triggers and anatomies of seismically detected rockfalls"

_Earth Surface Dynamics, 2017_

## Short Comment (SC1) · 27 May 2017

This manuscript describes seismic signals of rockfalls and discuss possible triggering mechanisms. The part about the detection, location and interpretation of rockfalls signals presents more detailed analysis and additional data compared to their previous study (Dietze et al. ESDD 2017). The second part is very speculative. The limited duration of the catalog and the limited number of events does not always support their interpretation. In addition, no uncertainties are given and no statistical tests are performed, so that it is not clear wether the results are statistical significant or are just due to random coincidences.

The two parts are rather disconnected: we don't need to analyze in details the seismic signals to understand their triggering mechanisms. In my opinion, it would have

been better to include some parts of the manuscript into the previous manuscript, and focus this second manuscript on the triggering mechanisms. As it is, the manuscript keeps moving from one topic (analysis of seismic signals) to the other one (triggering mechanisms). It makes the manuscript difficult to read and the message unclear.

1) Interpretation of seismic signals

Detection. p7 l5. I think the time criteria used to detect rockfalls (max duration and time delay between stations) are two severe and may reject true events (see review of previous paper)

p11 l27. Why not looking at the full catalog from the Swiss Seismological Service (not only d<20 km and m>1) in order to classify seismic events? I think that a significant fraction of your 359 events classified as earthquakes should be listed in this catalog, and the earthquake catalog could be useful for an automatic classification of events.

Rockfall volume. For events detected by TLS, could you add the estimated volume in the supplementary manuscript? For the other events, don't you have pictures of the cliff (before and after the monitoring period) that might allow you to detect the largest events and to estimate their volume?

Location. p13. Frequency range. How do you adjust the frequency range: manual or automatic? On which criteria? How do you know if a location is good or not (if you don't have TLS data)?

Figure 3. I do not understand the time interval you use to locate each pulse. Could you add this information in the table of Fig 3 (or in a separate table)? I am surprised that you can separate pulses so close in time looking at correlation between signal envelopes, and with only 4 stations. And that the results seem so accurate! It looks almost too nice to be true. Maybe add a plot showing the envelope of the filtered signal at each station (frequency range used for location), zoomed around time of pulses 0-3?

Organization of the manuscript. Why don't you present the interpretation of seismic

signals in section 4, just after the description, rather than in the discussion section 5? When reading 4.2, I was curious about the interpretation of these signals. But by the time I reached the discussion section, I had already forgotten what I read in section 4. The discussion should be more general, and compare your results with previous studies.

Classification of rockfalls I don't find the classification of events in type A (free fall), B (multiple impacts and C (avalanche) meaningful. One event can apparently belong to several classes, the event shown in Figure 2 is classified in type A (p20, l6), but the last part of this signal is of class C (p20, l11). Event III is a combination of types B and C (p22).

The first pulse of the signal is interpreted as an impact (e.g., p20 l18, l21, l24, p21 l15 ). Don't you think that it could be rather a detachment phase (elastic rebound)? Can you show the cliff profile at the location of a few events, and does the cliff geometry supports the hypothesis of an impact just below the detachment area? p20, l22: Vilajosana (2008) is not a good reference to discuss the detachment process, because this study describes an artificially triggered rockfall.

Location There is not enough information on the location method. p20, l24 : How do you adjust the time window to locate the beginning of the signal? Does your method works with short time windows, so that you can distinguish successive impacts? Don't you need time windows much longer than the time delay between stations (about 1 s)? p20, l26. I don't understand where this correction comes from?

p21 l29. One rockfall is located about 125 m above a preceding rockfall, that initiated 26 s earlier, and is supposed to have been triggered by the impacts of the preceding event. I find unlikely that a rockfall can be triggered above a first event 26s later, especially since the rockfall was probably quite small and the recorded ground velocity not very large (2 microm/s). Could it be a location error, so that the second rockfall initiated next to the previous one or close to an impact zone of the previous event?

5.1.3. I don't agree with the fall heights computed in this section and with the interpretation of the seismic signal. Maybe I missed some explanation or misunderstood something? p22, l17. A free fall height of h= 122m should last t=sqrt (2*H/g) = 4.98 s , not 3.52 s. For h=112, t=4.78s (not 3.38). And for H=795 m, t =12.7 s, not 6.9 s. The total fall time of 6.9 s should correspond to a fall height of 235 m for free fall, but the intermediate impact should have decreased the bloc velocity, so that the fall height should be smaller. This implies that the two phases could correspond to the initiation of the rockfall (phase 1) and to the impact at the base of the cliff (phase 2), rather than 2 distinct events.

p22 l24. I am not convinced that "seismology can provide insights . . . that not other method could achieve". A video camera would provide a more accurate description of a rockfall event (if it occurs during a sunny day).

2) Triggering mechanisms

- Temporal variation of rockfall elevation (Figure 6 and section 5.4) I am not convinced by the results shown in Figure 6 and by their interpretation.

First the figure 6 is hard to understand because the x axis is not continuous (events of 2014 are shown after events in 2015). Replacing the x axis by "day of year" would make this figure easier to understand. But if you believe those variations are due to seasonal effects, why don't you fit your data by a sinus function rather than by a straight line? A straight line would imply a jump at the end of each year ... this does not seem very physical.

Looking at figure 6, and based on other studies of the same cliff (Strunden et al 2014), I can believe that events in winter have a higher elevation than in summer. The proposed interpretation seems reasonable (the upper part of the cliff is more exposed to freeze-thaw cycles in winter than the lower part of the cliff which is the shade all day in winter).

But I remain doubtful about the proposed linear trend and its interpretation (p28 l10,

continuously lowering drying front along the cliff face). Precipitation should induced important and fast changes in the water table, and overwhelm the seasonal fluctuations. Since the number of events is quite small and the correlation rather weak, It is important to test statistically if this correlation is statistically significant and to provide uncertainties on the slope parameter.

Correlation with precipitation and temperature Because of the limited time range of your dataset and the limited number of events, it is crucial to test wether your results are statistically significant or could be due to random coincidence. One method would be to apply the same methods as in Figures 7 and 9 to a random catalog, obtained by assuming a uniform distribution of rockfalls in time during the monitoring period.

Influence of precipitations p24, l6. Why don't you cite here Helmstetter and Garambois (2010), who analyzed the influence of precipitation on rockfall activity, - several years before the two other cited references, - with a longer datased (3 yrs) - with more events (several thousands), - and a better temporal resolution (5 mn)? I usually avoid citing my own papers in a review, but here I could not resist.

Figure 8 I am not convinced by the diurnal variations of rockfall activity shown in Fig 8a. Such variations are similar to statistical fluctuations expected for a poisson process (uniform distribution in time). And the fact that there are slightly less events during the day may be explained by the increase in the seismic noise during the day, so that the smallest events are missed.

Conclusion. Why don't you decrease the sampling rate of your meteorological data to a few minutes? It seems much easier to me than trying to estimate precipitation from seismic data ...

Details

p2 l4 : "10 minutes" rather than "less than 20 h"

p2 l9. You should add a reference of Lacroix and Helmstetter (2011) as it was the first

study to locate rockfall seismic signals.

p2 l15. You should cite here Deparis et al (BSSA 2008) and Dammeier et al. (2011) instead of Helmstetter and Garambois 2010. Deparis et al (2008) and Dammeier et al. (2011) both used a national seismic network designed to detect earthquakes, while Helmstetter and Garambois (2010) used a seismic network devoted to the monitoring of the rockslide.

p2 l19. You should cite here D'amato et al (2016)

p2 l29. I don't understand "solution of solids". Do you mean "dissolution of solids"?

p2 l26. I don't understand the word "Anticipation" here. Maybe replace by "identification"?

p3 l14. I don't agree with the word "overwhelm" : Rockfalls can be triggered by earthquakes for shaking much smaller than gravitational acceleration (e.g., Meunier et al GRL 2007).

p3 l15. I am not sure that triggering by earthquakes is immediate, but there are few papers about time delay between earthquakes and triggered events. Lacroix et al (2015, doi:10.1016/j.rse.2015.05.010) suggest that there can be a delayed response of a landslide to a shaking.

p3 l23. Section 2.3 deals with precipitation rather than meteorological triggers, the effect of temperature is described in another section.

p9 l13. Why these thresholds of m=1 and d=20 km? Are there no larger and more distant earthquake that could have produced a stronger shaking?

p11 l7. Why not using UTC times? It would make the analysis much easier to have a continuous time.

p11 l15. You should explain why you detected 17 events in this manuscript for 2014 but only 10 in your previous manuscript.

p13 Figure caption. replace 2014 by 2015.

p28. l2: Do you mean "upper" instead of "lower"?

p28 l17. Rocks are not "mobile" before falling from the cliff. Rather replace by "loose"?

Suppl. Material : Can you add rockfall volume for events detected by TLS?

---

## Author Comment (AC1) · 4 Sep 2017

**Response to referee comments – referee 1**

[Spatiotemporal patterns, triggers and anatomies of seismically detected rockfalls]
August 30, 2017

We would like to thank the referee for the encouraging and helpful comments, all of them obviously devoted to improve the quality and impact of the manuscript.

**Referee 1.1**: *This manuscript describes seismic signals of rockfalls and discuss possible triggering mechanisms. The part about the detection, location and interpretation of rockfalls signals presents more detailed analysis and additional data compared to their previous study (Dietze et al. ESDD 2017). The second part is very speculative. The limited duration of the catalog and the limited number of events does not always support their interpretation. In addition, no uncertainties are given and no statistical tests are performed, so that it is not clear wether the results are statistical significant or are just due to random coincidences.*

**Reply**: To account for the identified shortcomings we do now provide the demanded uncertainty estimates where possible. The inferred trend of decreasing rockfall activity elevation with the time of the year is now supported by statistic tests and Monte Carlo-based estimates of uncertainty. The values of statistic tests for a potential Poisson process in the diurnal distribution of rockfalls is now also discussed in the text. Please also see our extended comments to point 1.21.

**Referee 1.2**: *The two parts are rather disconnected: we don't need to analyze in details the seismic signals to understand their triggering mechanisms. In my opinion, it would have been better to include some parts of the manuscript into the previous manuscript, and focus this second manuscript on the triggering mechanisms. As it is, the manuscript keeps moving from one topic (analysis of seismic signals) to the other one (triggering mechanisms). It makes the manuscript difficult to read and the message unclear.*

**Reply**: Indeed, the scope of the manuscript is not limited to the triggers of rock falls. We make full use of the detailed information available through the seismic approach. While Dietze et al. (2017 a) focused on comparing seismic results with independent TLS data to investigate the validity, precision and limitations of the seismic method, this current manuscript expands

the range of applications to show for the first time how one method integratively allows investigating what, when, how happens, which temporal and spatial patterns emerge at different scales of interest, and by which lag time rockfalls reacted to multiple external trigger activities. Thus, we see the manuscript presenting more than just rockfall trigger analysis after a signal processing/description part.

Alternatively, splitting the content into individual manuscripts appears too much a salami slice approach. The other alternative, moving the event characterisation part to the manuscript by Dietze et al. (2017 a), as suggested by the referee, or collapse both manuscripts (one option during the initial stage of the study) would have diffused the focus of Dietze et al. (2017 a) and caused a mis fit of the underlying data sets. The TLS control data was only available for about one month while seismic data was recorded for about half a year without hard TLS-based control data. Thus, argumentation for the validation part would have been at least cumbersome and somewhat unjustified.

However, since there evidently was confusion about the structure and organisation of the manuscript we expanded the introduction section to account for this shortcoming.
* * *
**Referee 1.3**: *Interpretation of seismic signals Detection. p7 l5. I think the time criteria used to detect rockfalls (max duration and time delay between stations) are two severe and may reject true events (see review of previous paper).*

**Reply**: Regarding the comments to the manuscript by Dietze et al. (2017 a) we may quote the referee: "Events detected at different stations are considered to be the same event if the time delay between stations is less than 1.75 s corresponding to an S wave with a velocity of 2000 m/s. I suggest increasing this value to about 10 s". Our reply to this comment holds here, as well: "location of the rockfall events in this study is only possible when the same seismic source (e.g., detachment process or impact) is recorded by all (at least four) stations. Allowing for larger time windows would indeed cause triggering of different event phases by different stations and thus, at best, a smearing of the location estimate. Thus, we need to keep this narrow time window". Accordingly, we now added an explaining sentence to the manuscript to solve this ambiguity.

We may further quote the referee: "Events longer than 20 s are removed because this is longer than the expected rockfall propagation. But rockfalls frequently occur in sequences of events, so that this constrain may remove true rockfall events". Our reply to this comment was: "Correct, rockfalls – also some of the events described in this manuscript – consist of sequences of activity, including talus slope mobilisation (e.g., event 8). However, the

constraint of 20 s is only used for the STA/LTA picker phase. Sequences of releases would result in several subsequent but short STA/LTA picks, as shown in figure 4b of the manuscript". In the context of this text here, the quote means that the 20 s constraint would only apply if all of the subsequent impact signals would last for more than 20 s, which is not the case. We also added explanations for this point to the revised version of the manuscript.
* * *
**Referee 1.4**: *p11 l27. Why not looking at the full catalog from the Swiss Seismological Service (not only d < 20 km and m > 1) in order to classify seismic events? I think that a significant fraction of your 359 events classified as earthquakes should be listed in this catalog, and the earthquake catalog could be useful for an automatic classification of events.*

**Reply**: The main purpose for using the catalogue was to check for earthquakes as a potential trigger of rockfalls, not to work towards a classification system. As explained in Dietze et al. (2017 a) the study area is not suitable for automated classification approaches, because the mobilised volumes are very small and seismic signals due to anthropogenic, fluvial and meteorological activity are prominent. The catalogue certainly contains several of the 359 events detected by our local network, but very small, local quakes may be missing although vital for investigating their role as rockfall triggers at the studied site. When utilising the full catalogue there certainly is a distance beyond which earthquakes will not have any meaningful effect on rockfall activity anymore (e.g., Jibson and Harp, 2012; Marc, 2016). Therefore, we believe it is best to remain with the local expression of earthquake activity as measured by our network to have a more direct estimate of local ground acceleration. This is now added to the manuscript.
* * *
**Referee 1.5**: *Rockfall volume. For events detected by TLS, could you add the estimated volume in the supplementary manuscript? For the other events, don't you have pictures of the cliff (before and after the monitoring period) that might allow you to detect the largest events and to estimate their volume?*

**Reply**: The volumes were added to the table in the supplementary materials. For the other events, there were no systematically collected images. The rockfall anticipated volumes were anyhow too small to be recorded by terrestrial photography. We would have to had taken drone-based images close to the cliff, a valuable idea though for future studies.
* * *
**Referee 1.6**: *Location. p13. Frequency range. How do you adjust the frequency range: manual or automatic? On which criteria? How do you*

*know if a location is good or not (if you don't have TLS data)?*

**Reply**: The frequencies were manually adjusted based on the spectra for each of the pulses. This information is now given in the manuscript. Indeed, the influence of the frequency window can be considerable when attempting to locate such short pulses, as discussed in Dietze et al. (2017 a). We also discuss this limitation already in the initial version of the manuscript (p. 21, l. 31 to p. 22, l. 2).
* * *
**Referee 1.7**: *Figure 3. I do not understand the time interval you use to locate each pulse. Could you add this information in the table of Fig 3 (or in a separate table)? I am surprised that you can separate pulses so close in time looking at correlation between signal envelopes, and with only 4 stations. And that the results seem so accurate! It looks almost too nice to be true. Maybe add a plot showing the envelope of the filtered signal at each station (frequency range used for location), zoomed around time of pulses 0–3?*

**Reply**: Event location was performed with exactly the values denoted in figure 3 d. We rephrased the figure caption slightly to make this clearer. Indeed, the time intervals are short. The location estimates appear so accurate because they were clipped at the 0.99 quantile for illustrative reasons, which is now added to the caption. We added the full R code used to process the data as well as plots of the spectra and the envelopes for all pulses and stations to the supplementary materials and added a short reference to these additional information to the manuscript.
* * *
**Referee 1.8**: *Organization of the manuscript. Why don't you present the interpretation of seismic signals in section 4, just after the description, rather than in the discussion section 5? When reading 4.2, I was curious about the interpretation of these signals. But by the time I reached the discussion section, I had already forgotten what I read in section 4.*

**Reply**: We followed the suggestion and interpret the signals just after describing them.
* * *
**Referee 1.9**: *The discussion should be more general, and compare your results with previous studies.*

**Reply**: We added discussions with respect to previous findings where useful. The entire discussion has been revised based on comments from all three referees.

**Referee 1.10**: *Classification of rockfalls I don't find the classification of events in type A (free fall), B (multiple impacts and C (avalanche) meaningful. One event can apparently belong to several classes, the event shown in Figure 2 is classified in type A (p20, l6), but the last part of this signal is of class C (p20, l11). Event III is a combination of types B and C (p22).*

**Reply**: The classification scheme has been revised, also in agreement with comments by referee 3.
* * *
**Referee 1.11**: *The first pulse of the signal is interpreted as an impact (e.g., p20 l18, l21, l24, p21 l15 ). Don't you think that it could be rather a detachment phase (elastic rebound)? Can you show the cliff profile at the location of a few events, and does the cliff geometry supports the hypothesis of an impact just below the detachment area? p20, l22: Vilajosana (2008) is not a good reference to discuss the detachment process, because this study describes an artificially triggered rockfall.*

**Reply**: We replaced the inadequate reference and rewrote the first misleading statement (p.20 l. 18 in the initial manuscript) by the more general term: "first clear rockfall related signal". At the end of the same paragraph we discuss that it is not clear from our data whether such a signal is related to an elastic rebound or a first impact for mobilised rock volumes predominantly below 1 m$^3$. The location estimates (based on a 10 m grid) for all detected events are not precise enough to test whether the cliff geometry would allow an intermediate rock mass impact a few m below a detachment spot. However, the general cliff slope (88.5 °) and the presence of several small ledges would in principle support this and the existence of type A (previously A) rockfall events also argue for intermediate impacts.
* * *
**Referee 1.12**: *Location There is not enough information on the location method. p20, l24 : How do you adjust the time window to locate the beginning of the signal? Does your method works with short time windows, so that you can distinguish successive impacts? Don't you need time windows much longer than the time delay between stations (about 1 s)?*

**Reply**: Part of the ambiguities came from the initial sentence structure. We did not explicitly attempt to change the time window to focus only on the first impact. Rather we used the STA/LTA pick, i.e., the entire picked event duration. The sentence has been rewritten. With respect to the second part of the referee comment, we describe the location approach in the methods (a buffer of 2 seconds is added to the STA/LTA pick). Additionally, we now exemplary show envelopes of discrete impact pulses and how they are offset

amongst the stations in the supplementary materials (see point 1.7). From these figures it is obvious how the cross correlation is possible also for such short time series, as long as the pulses are significantly shorter than the time series itself.
* * *
**Referee 1.12**: *p20, l26. I don't understand where this correction comes from?*

**Reply**: The correction is based on the time lags between seismic signals preceding the first clear impact signals, sometimes visible in seismic records, and the impact signal itself (mentioned at the end of chapter 4.2 in the initial manuscript version). We now repeat this result in where we convert it to free fall distances and believe the ambiguities are solved by this.
* * *
**Referee 1.13**: *p21 l29. One rockfall is located about 125 m above a preceding rockfall, that initiated 26 s earlier, and is supposed to have been triggered by the impacts of the preceding event. I find unlikely that a rockfall can be triggered above a first event 26 s later, especially since the rockfall was probably quite small and the recorded ground velocity not very large (2 microm/s). Could it be a location error, so that the second rockfall initiated next to the previous one or close to an impact zone of the previous event?*

**Reply**: Indeed, the event is of low seismic magnitude. But while the initiation of the first rockfall (phase 0 to 4 in figure 4) was about 26 seconds before the second rockfall (phase 5 in figure 4) the first rockfall lasted longer than the second one. Hence, ground excitation by the first event continued even after the second event already happened. Thus, the actual time lag between the two rockfalls is zero.

The location estimate polygons of phase 2 to 3 and phase 5 do overlap slightly and, obviously, the selected threshold quantile for clipping the polygons might not be appropriate. However, we also used other filter frequency windows and changed the signal onset and duration for the location approach (at least for phase 5, because phases 2 and 3 were too close to each other in time) but yielded similar location estimate polygons. We now discuss the alternative interpretation that the signal of phase 5 might also come from a rock mass being released from or near to the spot of the first one.
* * *
**Referee 1.15**: *5.1.3. I don't agree with the fall heights computed in this section and with the interpretation of the seismic signal. Maybe I missed some explanation or misunderstood something? p22, l17. A free fall height of h = 122 m should last t = sqrt (2 * H / g) = 4.98 s , not 3.52 s. For h =*

*112, t = 4.78 s (not 3.38). And for H = 795 m, t = 12.7 s, not 6.9 s. The total fall time of 6.9 s should correspond to a fall height of 235 m for free fall, but the intermediate impact should have decreased the bloc velocity, so that the fall height should be smaller. This implies that the two phases could correspond to the initiation of the rockfall (phase 1) and to the impact at the base of the cliff (phase 2), rather than 2 distinct events.*

**Reply**: Indeed, we recalculated the free fall distances now. The initial interpretation of the scenario has been adjusted accordingly.
* * *
**Referee 1.16***: p22 l24. I am not convinced that "seismology can provide insights ... that not other method could achieve". A video camera would provide a more accurate description of a rockfall event (if it occurs during a sunny day).*

**Reply**: We removed the inappropriate phrase and provide more concise information, now.
* * *
**Referee 1.17***: 2) Triggering mechanisms — Temporal variation of rockfall elevation (Figure 6 and section 5.4) I am not convinced by the results shown in Figure 6 and by their interpretation. First the figure 6 is hard to understand because the x axis is not continuous (events of 2014 are shown after events in 2015). Replacing the x axis by "day of year" would make this figure easier to understand. But if you believe those variations are due to seasonal effects, why don't you fit your data by a sinus function rather than by a straight line? A straight line would imply a jump at the end of each year ... this does not seem very physical.*

**Reply**: We removed the information about the year from the x axis of figure 6 c and added a secondary Julian day axis to the plot. We did however not fit a continuous function to the data because our interpretation of the data is that it does not follow a continuous, sinusoidal forcing mechanism like insolation or air temperature. Rather, we consider the "recharging period" (5.4) to be step-like, resetting the activity window back to the cliff top during winter time, from which it progresses down the cliff during the following spring and summer season. Thus, the fit function would take the shape of a reverse sawtooth wave. But since we have no data for the sharp ramp upwards we did not apply this function strictly. However, we follow the idea to use a more physically justified model and perform the fit now with an exponential model (cf. point 1.18).
* * *
**Referee 1.18***: Looking at figure 6, and based on other studies of the same*

*cliff (Strunden et al 2014), I can believe that events in winter have a higher elevation than in summer. The proposed interpretation seems reasonable (the upper part of the cliff is more exposed to freeze-thaw cycles in winter than the lower part of the cliff which is the shade all day in winter). But I remain doubtful about the proposed linear trend and its interpretation (p28 l10, continuously lowering drying front along the cliff face). Precipitation should induced important and fast changes in the water table, and overwhelm the seasonal fluctuations. Since the number of events is quite small and the correlation rather weak, It is important to test statistically if this correlation is statistically significant and to provide uncertainties on the slope parameter.*

**Reply**: The effect of precipitation can have (at least) two modes. Precipitation that falls directly onto the cliff face and precipitation that falls within the catchment above the cliff. The first mode will indeed cause rapid changes in the rock, but only if the rock is also able to dry up during periods of dry weather, without being fed with water from within bedrock. This will only be the case when the water table within the rock has already dropped below the altitude of interest. The other mode, precipitation falling within the catchment above the cliff, will have a different effect. First, a significant part of it will be channelised and flow out of the area of interest through the many creeks and water falls, thus not affecting geomorphic processes at the cliff. Second, the part that infiltrates may contribute to lifting the water table within the rock mass but the most important period for recharging this water table will be during the thawing and snow melt season, not during rain events occurring throughout the year. In fact, we see the overall trend of a lowering water table with the course of the year in the progressively lower outlets of water along the cliff as shown in the supplementary material.

A linear trend was the simplest model to fit, although even this already yielded a rank correlation coefficient of $\tau = -0.56$, $p = 5.0 \cdot 10^{-7}$ (the inappropriate Pearson's correlation coefficient would be $-0.77$, $p = 6.4 \cdot 10^{-9}$). Thus, we regard the trend not as spurious.

Indeed, an exponentially declining trend is more appropriate from a hydrological perspective. We changed the fit model and added a Monte Carlo-based estimate of its scatter. We also discuss the significance of the model in the revised manuscript.
* * *
**Referee 1.19**: *Correlation with precipitation and temperature Because of the limited time range of your dataset and the limited number of events, it is crucial to test wether your results are statistically significant or could be due to random coincidence. One method would be to apply the same methods as in Figures 7 and 9 to a random catalog, obtained by assuming a uniform distribution of rockfalls in time during the monitoring period.*

**Reply**: In agreement with the suggestions by referee 2 we provide now Monte Carlo-based kernel density estimates for lag times between precipitation events (new figure 7) to show that these lag time density estimates are clearly distinct. The precipitation-precipitation lag time is significantly longer than the precipitation-rockfall lag time, implying that the observed lag times are not an artifact. Likewise, we show freeze-thaw and thaw-free lag times, also with significantly longer lag times than when considering their relationship with rockfalls.
* * *
**Referee 1.20**: *Influence of precipitations p24, l6. Why don't you cite here Helmstetter and Garambois (2010), who analyzed the influence of precipitation on rockfall activity, – several years before the two other cited references, – with a longer datased (3 yrs) – with more events (several thousands), – and a better temporal resolution (5 mn)? I usually avoid citing my own papers in a review, but here I could not resist.*

**Reply**: Done as suggested.
* * *
**Referee 1.21**: *Figure 8 I am not convinced by the diurnal variations of rockfall activity shown in Fig 8a. Such variations are similar to statistical fluctuations expected for a poisson process (uniform distribution in time). And the fact that there are slightly less events during the day may be explained by the increase in the seismic noise during the day, so that the smallest events are missed.*

**Reply**: Indeed, testing if the distribution of the data is be due to a Poisson process would probably result in a "Yes". However, our approach is to mechanistically understand and explain the underlying processes that can trigger rockfall. And the time lag analysis obviously shows that there is a relationship between triggers and rockfalls (e.g., figure 7 or argumentation in chapter 4.5).

The general problem is that the mechanisms we suggest, temperature and temperature change rate, have a sinusoidal form and their combined action results in an almost flat probability density curve and thus in wide distribution in time of events sampled from this distribution. The following R code and plot panel illustrate the effect for different sample sizes (10, 40 and 500) with an imposed uniform scatter of $\pm$ 1 hour.

```
**set sample parameters**
N <- 10^4 # only used to generate the trigger functions
n <- c(10, 40, 500) # choice of sample size
s <- c(-1, 1) # amount of scatter
```

```
**define time vector**
t <- seq(from = 1,
         to = 24,
         length.out = N)

**define probability vector for trigger 1**
p1 <- abs(sin(seq(from = 0,
                  to = 2 * pi,
                  length.out = N)))

**define probability vector for trigger 2**
p2 <- abs(cos(seq(from = 0,
                  to = 2 * pi,
                  length.out = N)))

**define joint probability vector**
p3 <- p1 + p2

**normalise probability vectors**
p1 <- p1 / sum(p1)
p2 <- p2 / sum(p2)
p3 <- p3 / sum(p3)

**generate sample from population**
x <- lapply(X = n, FUN = function(n, t, p3, s) {

  ## subsample data set
  x <- sample(x = t,
              size = n,
              prob = p3)

  ## add scatter
  x <- x + runif(n = n,
                 min = s[1],
                 max = s[2])

  ## correct for 1-24 hour constraint
  x <- ifelse(x > 24, x - 24, x)
  x <- ifelse(x < 0, x + 24, x)
}, t, p3, s)

**plot density estimate from sample**
jpeg(filename = "~/Desktop/kde.jpg",
```

```
        width = 2500,
        height = 1200,
        res = 300)

**define plot layout**
plot_layout <- rbind(c(1, 1, 1),
                     c(2, 3, 4))

layout(mat = plot_layout)

**plot probability vectors**
plot(x = t,
     y = p1,
     type = "l",
     main = "Probability functions for cyclic drivers",
     xlab = "Hour of the day",
     ylab = "Density")

mtext(text = "Black = sin(), Red = cos(), Blue = sin() + cos()",
      side = 3,
      cex = 0.8)

lines(x = t,
      y = p2,
      col = 2)

lines(x = t,
      y = p3,
      col = 4)

**generate KDE plots**
lapply(X = x, FUN = function(x) {

  plot(density(x,
               bw = 1),
       xlim = c(0, 24),
       main = paste("KDE (n = ",
                    length(x),
                    " , kernel size = 1)",
                    sep = ""),
       xlab = "Hour of the day",
       ylab = "Density")
  rug(x = x)})
```

[Figure]

Figure 1: Kernel density estimates of synthetic events, based on a combined sine and cosine probability of occurrence. The three density estimates resulted from 10, 40 and 500 synthetic samples, respectively. The red curve represents diurnal temperature. The highest probabilities (i.e., the absolute values of the sine function) are during the coldes hours and the hottest hours of the day. The black curve depicts temperature change rate as first derivative of the red curve, whose representation as probabilites is again done by computing the absolute values. Superimposing both curves yields the combined probability, i.e., the blue line.

```
dev.off()
```

Apparently, the resulting density estimate curves always look like as being the result of a Poisson process, regardless of the sample size. Thus, we see limited new insight from investigating if the rockfall event distribution in the manuscript would pass or fail a Poisson distribution test. We discuss this general issue now in the text.

We consider the alternative explanation that detecting a slightly lower number of events during daytime (10 am to 6 pm) may be due to increased seismic noise that hide rockfall signals only partly valid. The most dominant sources of "noise" are precipitation events (cf. discussion in Dietze et al., 2017 a) and the train running along the cliff top. Both sources are not restricted to daytime; precipitation events also occurred over night and the train runs from 6 am to 9 pm.
* * *
**Referee 1.22**: *Conclusion. Why don't you decrease the sampling rate of your meteorological data to a few minutes? It seems much easier to me than trying to estimate precipitation from seismic data ...*

**Reply**: The data we used was only provided as hourly values. We did not record our own meteorological data. This is now crafted into the revised manuscript.
* * *
**Referee 1.23**: *p2 l4 : "10 minutes" rather than "less than 20 h"*

**Reply**: Our statement "a data base of 144 rockfalls with a time uncertainty of less than 20 h for some of the events" is a direct transcription of the information given by D'Amato et al. (2016), section 4.2: "Out of the 214 rockfalls forming DB2, we have studied 144 rockfalls, whose date is known with an uncertainty lower than 20 h". In fact, we cannot give more detailed information about the time resolution, since D'Amato et al. (2016) mention a series of complicating conditions (fog/precipition, snow, night time) that limit the temporal resolution of their data base.
* * *
**Referee 1.24**: *p2 l9. You should add a reference of Lacroix and Helmstetter (2011) as it was the first study to locate rockfall seismic signals.*

**Reply**: Done as suggested.
* * *
**Referee 1.25**: *p2 l15. You should cite here Deparis et al (BSSA 2008) and Dammeier et al. (2011) instead of Helmstetter and Garambois 2010. Deparis et al (2008) and Dammeier et al. (2011) both used a national seismic network designed to detect earthquakes, while Helmstetter and Garambois (2010) used a seismic network devoted to the monitoring of the rockslide.*

**Reply**: Done as suggested.
* * *
**Referee 1.26**: *p2 l19. You should cite here D'amato et al (2016)*

**Reply**: Done as suggested.
* * *
**Referee 1.27**: *p2 l29. I don't understand "solution of solids". Do you mean "dissolution of solids"?*

**Reply**: Changed as suggested.
* * *
**Referee 1.28**: *p2 l26. I don't understand the word "Anticipation" here. Maybe replace by "identification"?*

**Reply**: Section title has been revised.
* * *
**Referee 1.29**: *p3 l14. I don't agree with the word "overwhelm": Rockfalls can be triggered by earthquakes for shaking much smaller than gravitational acceleration (e.g., Meunier et al. GRL 2007).*

**Reply**: We removed the misleading sentence.
* * *
**Referee 1.30**: *p3 l15. I am not sure that triggering by earthquakes is immediate, but there are few papers about time delay between earthquakes and triggered events. Lacroix et al (2015, doi:10.1016/j.rse.2015.05.010) suggest that there can be a delayed response of a landslide to a shaking.*

**Reply**: Indeed, for landslides a time lag can be justified. For rockfalls, this is not so clear, at least to the extend we screened the available literature. We note this point now in the manuscript.
* * *
**Referee 1.31**: *p3 l23. Section 2.3 deals with precipitation rather than meteorological triggers, the effect of temperature is described in another section.*

**Reply**: The section also includes potential effects of wind and lightning strikes. Thus, we think "meteorological triggers" is an adequate description.
* * *
**Referee 1.32**: *p9 l13. Why these thresholds of m=1 and d=20 km? Are there no larger and more distant earthquake that could have produced a stronger shaking?*

**Reply**: These thresholds were a conservative approach to the question. Even for large landslides, activation will only happen within a few tens of kilometers if the magnitude is around 4 or greater (cf. Marc 2016, Marc 2017). This information has been added to the text, now.
* * *
**Referee 1.33**: *p11 l7. Why not using UTC times? It would make the analysis much easier to have a continuous time.*

**Reply**: The dates are only given in local time in the mansucript, the analysis was done using UTC time. We decided to use local time to better align with the anthropogenic/daytime cycle.
* * *
**Referee 1.34**: *p11 l15. You should explain why you detected 17 events in this manuscript for 2014 but only 10 in your previous manuscript.*

**Reply**: Information added as suggested.
* * *
**Referee 1.35**: *p13 Figure caption. Replace 2014 by 2015.*

**Reply**: Done as suggested.
* * *
**Referee 1.36**: *p28. l2: Do you mean "upper" instead of "lower"?*

**Reply**: Well, we mean middle and lower, actually. The upper sections are in the sun anyway. It is the successively lower sections that experience more and more sun shine hours.
* * *
**Referee 1.37**: *p28 l17. Rocks are not "mobile" before falling from the cliff. Rather replace by "loose"?*

**Reply**: Replaced as suggested.
* * *
**Referee 1.38**: *Suppl. Material : Can you add rockfall volume for events detected by TLS?*

**Reply**: Done as suggested.

**1 Cited literatue**

D'Amato, J., Hantz, D., Guerin, A., Jaboyedoff, M., Baillet, L., and Mariscal, A.: Influence of meteorological factors on rockfall occurrence in a middle mountain limestone cliff, Natural Hazards and Earth System Sciences, 16, 719735, doi:10.5194/nhess-16-719-2016, 2016.

Dietze, M., Mohadjer, S., Turowski, J. M., Ehlers, T. A., and Hovius, N.: Validity, precision and limitations of seismic rockfall monitoring, Earth Surf. Dynam. Discuss., in review, 2017 a.

Jibson, R. W., and Harp, E. L.: Extraordinary Distance Limits of Landslides Triggered by the 2011 Mineral, Virginia, Earthquake, Bulletin of the Seismological Society of America 102, 2368–2377.

Marc, O., Hovius, N., Meunier, P., Gorum, T., and Uchida, T. A seismologically consistent expression for the total area and volume of earthquaketriggered landsliding. JGR 121, 640–663, 2016.

ADD MARC 2016.

**Response to referee comments – referee 2**

[Spatiotemporal patterns, triggers and anatomies of seismically detected rockfalls]
August 30, 2017

We would like to thank the interactively commenting researcher for the encouraging and helpful comments, all of them obviously devoted to improve the quality and impact of the manuscript.
* * *
**Referee 2.1**: *The paper analyses a very useful rockfall inventory in order to identify triggers and corresponding time lags. But the analysis is not very convincing and should be more developed. Moreover, it's important to know the context in which the conclusions apply, notably the order of magnitude of the rockfall volumes and the geological structure of the cliff (thickness, dip, dip direction of the beds).*

**Reply**: As a consequence of the here presented comments as well as the comments by referee 1 we are confident that the justification, quality and robustness of the analysis has improved with the revised version of the manuscript, at least as the amount and nature of the data allows generating more robust results. Where this was not the case the manuscript now mentions the imitations with respect to the interpretation of the data and analysis results.

As a matter of the analysis technique (seismic monitoring of "very small" rockfall events) it is not possible to provide any reasonable estimates of the volumes of the detected rockfalls. The article by Dietze et al. (2017 a) discusses this issue at more detail. Principally, it is the interaction of mobilised rock masses with the cliff and the talus slope that precludes straightforward relations of seismic properties such as distance-corrected amplitude or energy integral with mobilised volumes. Thus, we now explicitly point at the small size of rockfalls in the Lauterbrunnen Valley at the end of the introduction and the study area description.

We also provide a short description of the geological situation and a reference to more information.
* * *
**Referee 2.2**: *Definition of rockfall Page 2, lines 27-33. This paragraph is not consistent for the following reasons: A falling rock volume is rarely fully isolated before its detachment; The relative separation of an "isolated" volume is the result of a geological process which acts over several million years; There is a contradiction between the sentences "its subsequent detachment by a release mechanism activated by a driving force" (line 28) and "The release mechanism is essentially a decrease in the stabilising forces" (line*

*30); Actually, the detachment can result from "decreasing material strength OR increasing stress" (line 33). So I suggest this section to be rewritten.*

**Reply**: We rewrote the section.
* * *
**Referee 2.3***: Section 2 (Anticipation of rockfall triggers) General comment: The different processes described in the section can be triggers (= near immediate response) but also have a delayed action (for example, an earthquake can induce new cracks whose propagation until failure can take a long time). So it would be proper to replace "The reaction of a rock mass to excitation by an earthquake is almost immediate" (page 3, line 15) by "The reaction of a rock mass to excitation by an earthquake can be immediate".*

**Reply**: We added a general explaining sentence before listing the triggers. This sentence underlines that we only discuss the trigger role of processes, not their long-term effects. Thus, we believe that the sentence as stated in the initial manuscript version holds, especially since it ends with the word "trigger".
* * *
**Referee 2.4***: In the same way, I suggest to replace "is supposed to be immediate" (line 22) by "can be immediate".*

**Reply**: We added the term "trigger role" here, as well, to be clear that we do not refer to the long-term action of mass wasting processes.
* * *
**Referee 2.5***: Page 3, line 25. How precipitation can reduce pore pressure? It is usually assumed that precipitation increases pore pressure.*

**Reply**: We corrected the term. Our view on pore pressure is and was that the pore pressure is negative and this "negative value" is reduced towards zero, a clearly misleading approach.
* * *
**Referee 2.6***: Page 3, line 27: "Increasing the load requires time for rain water infiltration, percolation and retention inside the rockmass": Water doesn't need to infiltrate inside the rockmass for its weight to overhelm the superficial rock volumes which are prone to fall.*

**Reply**: We added information to be more concise. Specifically, to increase the load beyond water running along the surface, infiltration is necessary.
* * *
**Referee 2.7**: *Section 2.4 (page 4). Heat related triggers. It would be more proper to describe the freezing process before the thawing one.*

**Reply**: Done as suggested.
* * *
**Referee 2.8**: *I don't understand the meaning of "transgressive" (line 16).*

**Reply**: The misleading sentence was removed.
* * *
**Referee 2.9**: *Section 3 (Materials and methods) 3.1 (Study area) As the influence of meteorological factors will be studied in the paper, the climatic context of the area should be presented: Minimal and maximal temperatures, annual precipitation at the weather station (elevation of the station); Minimal and maximal elevations of the cliff; Temperature and precipitation gradient between the station and the base of the cliff. Remark: from the 1/50000 topographic map, the maximal height of the cliff would be 600-700 m and not 1000 m (between 1000 and 1600 m).*

**Reply**: Meteorological information has been added to the degree data could be identified. The valley height of about 1000 m refers to the entire valley (both sides and along the entire upper part). We now explicitly mention the drop in altitude for the valley section that has been studied.
* * *
**Referee 2.10**: *Page 8, line 32: "Snow melt-generated water input is regarded as irrelevant because the cliff face is snow free in winter due to the steep gradient." This sentence is contradictory with the following one (page 28, line 11): "Water storage is also refreshed by snow melting higher up in the catchment, which provides a more or less continuous supply of water that can seep into the karstic limestone plateau on top of the cliff during the melt season." This process has not been analyzed quantitatively, but its influence seems to be not negligible. I suggest to reformulate this section.*

**Reply**: Indeed, the two modes of snow-melt generated water were inappropriately expressed. The first one is related to water affecting the cliff face "from the outside" directly. The second one is related to a water table within the limestone cliff, affecting the cliff face "from the inside". The sentence has been rewritten.
* * *
**Referee 2.11**: *Page 9, line 15. I suppose that lag times for precipitation were defined as the time span between THE BEGINNING of a precipitation event with > 0.1 mm/h and the next rockfall, but it's better to precise.*

**Reply**: No, the lag times are defined as the time between the end of a precipitation event and the next occurring rockfall. If a rockfall occurs during a precipitation event, the time lag is zero. We now mention in the text that we refer to the end of a precipitation event.
* * *
**Referee 2.12**: *Section 4 (Results) Figure 5-a and b. The legend indicates a cumulative probability, but it seems to be rather a cumulative number (a probability should be lower than 1)*

**Reply**: The legend has been corrected. The correct term is "Cumulative normalised probabilities". Each rockfall location procedure yields a grid-based polygon with the probability of its occurrence assigned. The summation of all grid values results in the graphics shown in figure 5 a and b.
* * *
**Referee 2.13**: *Figure 6-a and b. The title of the vertical axis is incomplete: It should be precised "Number of events per week or per n days"; Moreover, the time interval is different between a and b. In March 2015 and in July and August 2014, the circles are not aligned with the rugs.*

**Reply**: The plots to which the axes refer are the histograms in figure 6 a and b. The bin widths were computed following the default function argument settings in R, which also explains the different time interval for a and b. Setting the bin widths to equal intervals would introduce a bias in the representation of the data. The number of samples in sub group 2015 is much higher than in sub group 2014. We added the bin widths to the axes labels for quantification issues. Circles and rugs were aligned.
* * *
**Referee 2.14**: *The text (page 18, lines 1-2) is not in agreement with Figure 6-a. From the figure (and contrary to the text), the three periods of enhanced activity occurred in the beginning of March (7 events), end of March (6 events), and beginning of April (7 events), with few activity after April 17. Figure 6a should be dilated horizontally for a better readability.*

**Reply**: The axes labels of a and b were twisted and have been corrected, now. Dilating panel a would come at the cost of readability of b and further break the common temporal x axis scale, relevant for having a first order look at this different rockfall activities.
* * *
**Referee 2.15**: *Page 18, lines 11-12 and Figure 7a. The density estimate of earthquake lag times (between an earthquake and the next rockfall) is*

*compared to the lag time between earthquakes. Uncertainty for the first one is visualized on Figure 7a, but what about the uncertainty for the second one?*

**Reply**: We now also present the uncertainty in earthquake-earthquake time lags based on the same Monte Carlo approach. This is also plotted now for precipitation and freeze-thaw/thaw-freeze transitions.
* * *
**Referee 2.16***: Could you give the mean time between earthquakes in the text (line 12)?*

**Reply**: We provide quartile ranges and the median to characterise lag times for this and the other lag time results in figure 7.
* * *
**Referee 2.17***: Page 18, lines 19-20 and Figure 7b. As for earthquakes, the lag time density estimates for precipitation events should be compared to the lag time between precipitations.*

**Reply**: We provide this visualisation, now.
* * *
**Referee 2.18***: Given that there are 108 rainfall events (page 24, line 10) for a monitoring period of 190 days, the mean lag time between precipitations is 42 hours. Assuming the precipitations and the rockfalls are regularly distributed in time, the mean lag time between precipitations and rockfalls should be 21 hours, which is dramatically longer than the observed peak of 1 hour. Of course, this oversimplified approach is not sufficient and the density function of the lag time between precipitations should be estimated as for earthquakes.*

**Reply**: Indeed, when using the real data the dry periods between precipitation events is lower than 21 hours, but still higher than the lag time between precipitation and rockfall. We show and discuss this point now in the manuscript.
* * *
**Referee 2.19***: Remark: The peak of the lag time density function is not the best parameter to characterize this function because it is not very marked. For a Poisson occurrence law (which could be a good model), the density function of the time between the events follows an exponential law, and the mode equals zero! So I suggest to indicate the mean or the median value. Assuming the rainfall events occur according to a Poisson law, the mean lag time would be 42 hours and the median lag time (mean lag time * ln2) 29*

[Figure]

Figure 1: Kernel density estimates of the precipitation lag times. Time scale linear (not logarithmic as in the manuscript). Time axis is clipped at 150 hours (full range of lag times goes to 387, i.e., 16.1 days). Vertical lines denote time lags of interest.

*hours. Assuming the rockfalls are uniformly distributed in time, the lag time between precipitations and rockfalls should be on average half of the lag time between precipitations. As the median lag time between precipitations and rockfalls is longer than 24 hours (Figure 7b), it seems that the hypothesis of a lag time of only several hours (anticipated in section 3, page 3, line 30) can be rejected and that the rockfalls doesn't occur preferentially after a rainfall.*

**Reply**: We understand the issue from a statistic point of view. However, from a physical and geomorphic point of view there is clear reason and empiric evidence that rockfalls, like other mass wasting processes, are strongly coupled to precipitation. We actually see that about half of all the rockfalls in this study occur within one day after a precipitation event, or, evidently most likely within less than four hours (cf. figure 7 in the manuscript and figure 1 in this text).

Our goal is not to describe the full density distribution function because, arguably, there is little reason to believe precipitation shall influence rockfall activity after, say, some hundred hours or more. Likewise, we see that the rockfalls are not uniformly distributed in time. They occur clustered (see figure 6, with the corrected time axis in the revised manuscript version), as we point out in chapter 4.4. Thus, we prefer to keep our characterisation of

the peak of the density curve. Perhaps the notion that the peak is "not very marked" is due to the logarithmic time scale in figure 7. However, looking at figure 1 in this text most likely shows that the peak is indeed marked. We mention this issue now in the revised manuscript, along with the notion that describing the peak of the density function is not aimed at giving a full description of the distribution but rather a description focused on the time scales relevant for rockfall, as also summarised in chapter 2.
* * *
**Referee 2.20**: *Figure 7e. Please explain what does mean "normalised" regression slope. Please give the percentiles which correspond to the different values given in the figure. Figures 7a-7e should be dilated horizontally for a better readability.*

**Reply**: The word "normalised" referred to the normalisation of the air temperature values prior to the regression analysis. The term has been removed from the y-axis label and the correct information is now given in the manuscript text and figure caption. Percentile (quartile) values are now given in figure 7 e. The entire figure will be larger once the manuscript is typeset. The smaller size is due to the online view constraints. The final size will be set by the journal regulations and is supposed to have a width of 8.3 cm instead of now 4.5 cm.
* * *
**Referee 2.21**: *Section 5 (Discussion) Page 22, line 32. "Within each of the activity periods the impacts are predominantly laterally spread by more than a kilometer". It's better to be coherent with page 18, line 4: "events close in time were separated by several hundred metres".*

**Reply**: The term on page 22 has been corrected.
* * *
**Referee 2.22**: *Page 24, line 6. My former comments on page 18, lines 11-12 and Figure 7a, applies also to this section. The analysis should be more advanced.*

**Reply**: Please see our reply to point 2.19.
* * *
**Referee 2.23**: *Page 24, line 13. the two prominent rockfall episodes in late May and late April were not associated with any rainfall. On Figure 6, there is no prominent rockfall episode in late May. On Figure 6a, there is a rainfall episode in late April! The section 5.3.2 should be more advanced.*

**Reply**: The ambiguities resulted from the twisted time axes in the original figure 6 a and b. Please see our corrections as discussed under point 2.14.

**Referee 2.24**: *Page 24, line 16. It seems on Figure 8 that the variability of weather-sensitive events is not very different from the one of the weather-insensitive events. So I would say that the peaks at 8 am and 8 pm are not significant (unless a statistical test prove the opposite).*

**Reply**: Again, from a statistical point of view one could easily argue this way. However, from a mechanistic perspective – explicitly accounting for the well-known trigger effect of precipitation – we are able to separate the diurnal rockfall occurrence distribution into events that show a lag time to precipitation and events that do not show such a link. Please see our argumentation and synthetic data example in point 1.21 from the first referee comments. We are not the first ones to identify rainfall as an important trigger for mass wasting processes, though not the only one.
* * *
**Referee 2.25**: *Page 24, line 18. The definition of precipitation-related rockfalls appears to be subjective. Comparing the time series of weather insensitive and precipitation-related rockfalls in Figure 8d, it appears there are not so different (clusters exist in both series). I suggest to test this temporal distributions against a Poisson law. If the first series is compatible with a Poisson process and not the second one, it would prove that the occurrence of these rockfalls is not aleatory and may be correlated with rainfall.*

**Reply**: The definition of precipitation-related rockfalls is based on the shape of the precipitation lag time density curve (figure 7 b in the manuscript and figure 1 in this text), which drops to almost zero within the first six hours. Thus, and even more pessimistically we set the threshold for precipitation-controlled events to four hours. One may say that this still is subjective, but at least it is based on observed data and a well-known relationship between trigger (precipitation) and effect (rockfall), as shown by many other investigations. Obviously, as long as the relationship can only be made based on the temporal connection one must rely on the existence of a physical mechanism that links the two processes.

We do not expect that the distributions of precipitation-related rockfall occurrence and weather-independent rockfall occurrence shall be fundamentally different from each other. On a diurnal basis there is little reason for that. Rather remarkably, the clusters for the precipitation-related events (blue dots in figure 8 d) are generally in agreement with the diurnal precipitation distribution (figure 8 c). From the other perspective, one could ask: Why would we expect rockfall events to occur randomly in time although a rich body of literature generated numerous arguments for the action of triggers that are, in turn controlled by environmental and meteorological conditions? Would it, from this perspective, not be a wrong assumption to

test for random event distribution?

Regarding the suggested test against a Poisson process, again, please see our discussion related to point 1.21 of the comments by referee 1.
* * *
**Referee 2.26**: *Page 24, line 23-29. In my opinion, the proposed interpretation is not convincing. Again, this interpretation needs to prove that the temporal distribution of the weather-insensitive rockfalls is not aleatory. A 2-3 h heat diffusion time lag is invoked to explain the delay for the group 8-11 am, but the group 0-6 am occurs before the minimal temperature. Why a heat diffusion time lag is not needed for this group? Why don't the warmest hours of the day cause stresses?*

**Reply**: For the first part of the comment, please see our arguments at point 2.24 and 2.25. For the second part, the 1–3 hour lag time section was a fragment and has been removed. It is indeed not needed to explain the patterns in the data. Witout it we see the following phases: i) 0–5 am (if excluding the one event around 0 am, the period shrinks to 2–5 am), which still experiences the coldest hours of the day. ii) 8–11 am, the strongest temperature increase rates, peaking between 7 and 9 am in spring and around 9 in summer/autumn. iii) 3–5 pm, strongest temperature decrease rates.

We added to the manuscript that this interpretation of the data is one out of more possibilities, though in our view a very plausible one as it accounts for the two first order effects of thermal forcing.
* * *
**Referee 2.27**: *Page 26, line 1-4 and Figure 9. The percentages given are not objective because: a) They depend on the choice of the maximal time lag (here 385 h); b) Only the freeze-thaw season must be considered for freeze-thaw-related rockfalls (it would increase the percentage). From a risk point of view, it would be more interesting to determine the rockfall frequency within one day after a precipitation or freeze-thaw event, and to compare it with the mean rockfall frequency which is 49/190 = 0.26 rockfall/day. The number of rockfalls occurred within one day after a rainfall is about 25 (Figure 9) and the number of rainfall events is 108, which gives a frequency of 25/108 = 0.23. So the risk is not increased by rainfall. This frequency seems to be higher for freeze-thaw events: 5/10 = 0.5.*

**Reply**: The "truncation" to 385 h (16 days) included all precipitation events. Thus, we cannot see any bias for the precipitation event statistics. Regarding the freeze-thaw events, one might argue if a maximum lag time of 16 days grades too much into the "not freeze-thaw affected" period of the year or not, i.e., to which extent argument b) from above applies or not.

Anyway, we removed the figure and interpretation from the manuscript because it evidently shifted the discussion too much in the the risk analysis scope, which is not what we intend for the article in this journal.
* * *
**Referee 2.28**: *Page 26, line 12. This value of ground acceleration is to compare with the accelerations of the earthquakes observed.*

**Reply**: Added as suggested.

**1 Cited literatue**

Dietze, M., Mohadjer, S., Turowski, J. M., Ehlers, T. A., and Hovius, N.: Validity, precision and limitations of seismic rockfall monitoring, Earth Surf. Dynam. Discuss., in review, 2017 a.

**Response to referee comments – referee 3**

[Spatiotemporal patterns, triggers and anatomies of seismically detected rockfalls]
August 30, 2017

We would like to thank the interactively commenting researcher for the encouraging and helpful comments, all of them obviously devoted to improve the quality and impact of the manuscript.
* * *
**Referee 3.1**: *In "Spatiotemporal patterns and triggers of seismically detected rockfalls", Dietze et al. exploit an array of broadband seismometers to locate individual rockfall events occurring across a cliff face in the Lauterbrunnen Valley, in the Swiss Alps during two periods – autumn 2014 and spring 2015. They subsequently assess the spatio-temporal evolution of rockfalls at this site and compare the located rockfall events against local meteorological data to make inferences regarding their trigger mechanisms. The first component of the study – the detection and location of rockfall events – is very nicely presented and provides convincing evidence regarding the exciting potential for seismic monitoring to help unravel the complex spatiotemporal evolution of geomorphic activity. I think that this on its own would be an interesting contribution that would fit very well with the broader remit of Earth Surface Dynamics. However, the subsequent analysis attempting to attribute the spatio-temporal patterns observed to different triggers and seasonality suffers from some serious flaws, leading to conclusions that are greatly overstated. I detail these issues below, in addition to a series of more minor revisions.*

**Reply**: In agreement with many of the points from referees 1 and 2, we revised the entire manuscript, especially the part focusing on the trigger patterns. For the term seasonality please see point 3.3. Regarding other statements that might weaken the conclusions drawn from our results, we added the limitations of the method and the subjectiveness of interpretations, where needed. Please see the individual points of the other and this referee letter.
* * *
**Referee 3.2**: *1. I would argue that you detect two, not three types of rockfall based on the seismic signals. My interpretation of the seismic signals presented is that there are two types of rockfall signal: (i) abrupt impulsive collisions of falling rocks against another surface, and (ii) the gradual acceleration and deceleration of rock avalanches. The other type described by the authors – multiple impulsive collisions – is really just a case of several*

*repeated instances of the former. Adding this as a separate third category is a bit misleading because it moves beyond the characteristics of the seismic signal and requires an additional layer interpretation based on the perceived likelihood that events are directly connected. In contrast, where you have Type A (or B) events triggering a Type C event, you do not make such a distinction although this would be equally valid as a classification as the suggested Type B event. I would suggest simplifying to two types of rockfall detected, and then make subsequent interpretations as required. This does nothing to diminish the significance of the paper, but separates out differences that are observed from interpretation of the temporal clustering of signals.*

**Reply**: The manuscript has been revised according to the suggestions.
* * *
**Referee 3.3**: *2. It is impossible to make robust inferences regarding seasonality based on the observations available. The reported observations span two monitoring periods that together capture less than 12 months. Given the stochastic nature of rockfall events, it is simply not possible to make inferences regarding the seasonal controls on rockfall hazards with any degree of confidence. One aspect that is picked out by the authors is the reduction in elevation of rockfall events between the spring and fall monitoring periods. However, this neglects the fact that the rockfall events are not only distributed vertically, but also laterally. The rockfall hotspots located within the two periods cluster in different parts of the cliff face. These also happen to be at slightly different elevations, but the most parsimonious explanation here is that the detected rockfalls are clustering around independent failure-prone areas and that the elevations are incidental. Multiple years of monitoring would be required before seasonality impacts can be reliably inferred. The fact that rockfall foci shift in space over time is interesting, and highlights the capacity for seismic arrays to monitor this process, but the authors should avoid over-reaching the limitations imposed by such a small monitoring period (it would, however, provide good motivation for a longer term study).*

**Reply**: We do not argue about seasonality but the seasonal scale. Both terms are now briefly defined to avoid confusion: "...the seasonal scale. In this scope, seasonal scale is a scale that focuses on the evolution of patterns over several months. It may not be mixed with the term seasonality, which would focus on the properties and dynamics of such patterns over a period of many repeated seasonal cycles".

Although there are activity hotspots the occurrence of rockfalls spreads over significant sections of the monitored cliff part. Rockfalls outside the clusters amount to 50 % in 2014 and 30 % in 2015. Still, even if rockfalls would cluster in different parts of the cliff there is still a significant exponentially decreasing trend of elevation with time of the year. Rockfall activity switches from one hotspot to another, and also outside of designated hotspots. Please also see point 3.50 for details.

The case of multiple years of monitoring already exists for this study area. The data of Strunden et al. (2014) already points at the existence of the trend though it suffers from the coarse temporal resolution. See our discussion in chapter 5.2.
* * *
**Referee 3.4**: *3. Ambiguity as to exactly how "trigger events" are defined. A major component of the study is to attribute detected rockfalls to specific "trigger events", predominately relating to meteorology, and the calculation of lag time separating triggers from a given rockfall. However, there are a number of issues to be addressed here. Firstly it is not clear from the authors' explanation exactly how trigger events are defined for all the meteorological variables assessed. This is particularly the case for wind, freeze-thaw and thermal gradients, for which an individual "trigger event" is conceptually more difficult to interpret as a single event - many of these processes would be likely to induce failure through repeated exposure and gradual weakening. The relevant methods section (Section 3.4) needs rewriting to make the rationale that determines the timing of each potential "trigger event" as clear as possible. Secondly, the authors acknowledge early on that triggers are not mutually exclusive, but are frequently additive. However they do not factor this into their analytical framework. This should be outlined towards the tail end of Section 3.4, prior to presenting and discussing the results. Without a more detailed assessment of how different processes interact, I am concerned that too much confidence is being given to the trigger attribution.*

**Reply**: Trigger mechanisms were assigned for all events based on the time lag between trigger action (precipitation event end, freeze-thaw transition of air temperature) and comparison of wind speed distributions during rockfalls with randomly sampled wind speed distributions. The temperature forcing is approached by regression slopes of air temperatures 3, 6 and 12 hours before a rockfall. All approaches are defined in chapter 3.4. To avoid confusion, these sub chapters were revised. The effect of wind speed and direction is now discussed in chapter 5.3.2.

We now explicitly mention that many of the triggers discussed in chapter 2 can also have a preparation effect but that we only focus on the trigger role. Thus, in the case of a mechanism repeatedly exciting a rock mass it would be the final excitation that matters, not the others before.

The interaction and superposition of trigger mechanisms has been addressed by checking if any of the events showed a meaningful process-relevant lag time for more than one trigger (e.g., if a precipitation event also showed

a close freeze-thaw transition, which is the case three times, as shown in figure 8d and already discussed in chapter 5.3.1). We now explicitly mention that we checked these time based co-occurrences in chapter
* * *
**Referee 3.5**: *4. Is wind likely to be uniform in complex terrain? It may still be that wind is not that important, but my expectation would be to see significant variations in wind speed across the site compared to the met station dependent on height within the valley, proximity to sheltering promontories in the cliff, and the wind direction. It is not clear whether the authors have attempted to account for this.*

**Reply**: Indeed, wind speed and direction distribution is likely to be far from uniform across the monitored cliff. We now mention this drawback of having data from only one weather station in chapter 3.4.3. Measuring spatially resolved wind speed and/or direction, especially in almost vertical alpine rock walls would to be a challenging idea.

We did not account for wind direction in our analysis, first because we did not include this measurement variable and second, though retrospective, because the wind speed distribution during rockfall events is so weak (about 4 m/s on average, cf. chapter 5.3.2).
* * *
**Referee 3.6**: *1. How reliable is detection of detachment? A number of references are made to locations of detachment and/or distances travelled between detachment and subsequent contacts. However, do all detachment events produce a signal that can be reliably detected?*

**Reply**: Well, actually we cannot provide any independent data on either location, exact time, or even the detachment process itself. Thus, seismic detection of detachment in terms of measuring the elastic rebound of the cliff and/or an impact of the rock mass just after it starts falling cannot be deciphered and confirmed in our case, at all. However, there are some other studies that present information for seismic signals of rock detachments (added to the revised manuscript, see also point 11 of referee 1). The entire section, interpreting the geoscientific meaning of such signals has been rewritten.

Regarding the second request, if all detachment events produce a signal that can be detected: No, this is not the case, as we already expressed in the initial manuscript version (p. 12, l. 6–8). We now also mention the number of observed cases with seismic signals prior to the first strong impact (12) when we interpret its role in chapter 5.1.
* * *
**Referee 3.7**: *2. Please avoid the "jet" colour scheme. The "jet" colour scheme suffers from a number of issues, the most important of which is that it is not perceptually uniform. One effect of this is that the jet colour map produces perceived sharp transitions where there are none. It is also difficult to interpret for people who are colour blind, and if printed in black and white. There are plenty of other alternatives that are perceptually uniform. Please use one of these instead.*

**Reply**: We thank the referee for the suggestion but this colour scheme is the quasi standard for showing spectrograms. We are aware of its shortcomings. During the development of the R package used to handle all the data and generating the plots we also experimented with other colour ramps but found that none of these alternative ones is as appropriate to depict the details inherent to the data sets as the one used in the manuscript. We use the spectrograms only for qualitative display of seismic signal properties. Thus, potential transition artifacts are not quantified in their interpretation. Readers are encouraged to work with the attached raw data. The R package eseis requires one to run no more than four functions to generate very customised spectrograms if needed.
* * *
**Referee 3.8**: *Page 1 Line 2: "Rockfalls are an essential geomorphic process" Odd choice of word "essential" – consider revising*

**Reply**: Replaced by "important".
* * *
**Referee 3.9**: *Page 1 Line 5: "independent information" – please be more specific so that it is clear exactly what you have done (i.e. compare against meteorological data)*

**Reply**: Additional information provided, as suggested.
* * *
**Referee 3.10**: *Page 1 Line 6: I would suggest that "ii) identify seasonally changing activity hotspots" is actually a subcategory of the following point: "iii) explore temporal activity patterns at different scales. . .". Please revise accordingly*

**Reply**: We changed the term to "identify spatial changes in activity hotspots" to be clear that the focus there is on the spatial domain. We believe this makes a sufficient distinction between i) event anatomy, ii) space and iii) time.
* * *
**Referee 3.11**: *Page 1 Line 20:". . .essential questions." Essential for what? Phrasing is a little awkward*

**Reply**: More details are given ("... essential across scientific disciplines").
* * *
**Referee 3.12**: *Page 2 Line 20: environmental seismology" is a very vague term – can you refine to something more specific to the methods employed?*

**Reply**: Does not the rest of the sentence ("the study of the seismic signals emitted by Earth surface processes") give a concise definition of 'environmental seismology? Larose et al. (2015) give a similar definition in their recent review article: "Environmental seismology consists in studying the mechanical vibrations that originate from, or that have been affected by external causes, that is to say causes outside the solid Earth". And during the first international conference/workshop on environmental seismology in July 2017 (EnviroSeis) the overall outcome also agreed on such a definition. Environmental seismology is field with a very diverse method portfolio and we believe it does little help here to open this entire portfolio in a section setting the scope of the manuscript.
* * *
**Referee 3.13**: *Anticipation of rockfall triggers. This section can be amalgamated into the introduction following the suggested changes:*

**Reply**: In an earlier version of the manuscript we had the anticipation section included in the introduction. However, as the introduction should be devoted to set the scope and justification for the manuscript content and the anticipation section contains both, a short review of important rockfall triggers and a conceptual formulation of what we finally anticipate and analyse, we agreed that it would distract the core role of an introduction section.
* * *
**Referee 3.14**: *Page 3 Lines 5-8. Remove all text from "In the following paragraphs. . ." onwards*

**Reply**: We rephrased the mentioned section to avoid summarising following content and clarify the role this and the following sections serve in the scope of the manuscript.
* * *
**Referee 3.15**: *Sections 2.1-2.5 are long and repetitive. This can be readily and succinctly summarised in a table (e.g. trigger, description, predicted lag time, references), which would be much more useful for reference. This table could be subdivided into the sections identified by the authors.*

**Reply**: We cannot see where the repetitiveness lies. Obviously, each sub-section follows a predefined logic. And reorganising this content in the form of a table would merely convert sentences into bullet points that eventually disguise vital information or pretend to quantify/classify information that should not be quantified/classified.
* * *
**Referee 3.61**: *Page 5 Line 26: "resampled to 10 m grid size". Please provide a reason to justify choosing to resample to a coarser grid*

**Reply**: Justification given as suggested.
* * *
**Referee 3.17**: *General comment: It would be good to have more details regarding the LiDAR scan (e.g. spatial area covered, fraction of area sampled at 1m grid resolution, point density) – it's a bit sparse at the moment.*

**Reply**: Details are provided, now.
* * *
**Referee 3.18**: *Page 6 Figure 1: Note that the map does not print very well in grayscale. Even in colour, the yellow text is difficult to read Section 3.3: Consider breaking down into sub-sections i.e. "detection", "classification", "source location" to help the reader navigate*

**Reply**: Figure has been revised to also plot adequately in grey scales. Subsections were introduced as suggested.
* * *
**Referee 3.19**: *Page 6 Line 6: "For this, the hourly raw signal files from both monitoring campaigns were appended to 25 hour long traces, overlapping by one hour". Could be clearer – presumably you mean that the data was collated into daily traces with a one hour overlap?*

**Reply**: Changed as suggested.
* * *
**Referee 3.20**: *Page 7 Line 5: Provide justification for choice of STA/LTA parameters.*

**Reply**: Justifications and references provided.
* * *
**Referee 3.21**: *Page 7 Line 7: Again need justification for threshold signal:noise ratio*

**Reply**: Justification provided. There and as an introducing statement at the beginning of the subsection, concerning all other parameters.
* * *
**Referee 3.22**: *Page 7 Lines 21-22: "as for example compiled by Burtin et al. (2016)" – shorten to: (Burtin et al., 2016)*

**Reply**: Done as suggested.
* * *
**Referee 3.23**: *Page 7 Line 22: "All remaining potential rockfall events were manually checked for agreement with these patterns" Are you excluding signals that don't look like the 10 previously recorded Lauterbrunnen rockfalls? What about signals that are similar to other previously published rockfalls and avalanches? Needs clarifying*

**Reply**: Clarified as suggested. We explain now that we checked that the signals were checked for distinctiveness from the other mentioned processes.
* * *
**Referee 3.24**: *General comment: throughout this section, reliant on citation "(cf. Dietze et al., 2017)" on several occasions. Would be good to diversify the references to acknowledge earlier work, especially given that this is still under peer review.*

**Reply**: References were removed whenever possible. However, since Dietze et al. (2017 a) actually provide the fundamental parameters for the seismic approach (picker setup, wave velocity estimate, location confidence intervals, nature of seismic signals other than rockfall), all of which are tailored to this study area, there is limited potential to cite different references.
* * *
**Referee 3.25**: *Page 8 Line 4: no need to italicise "m / s"*

**Reply**: Changed as suggested, also unit format aligned to journal guidelines.
* * *
**Referee 3.26**: *Section 3.4: Before talking about lag times, need to specify how the timing of each "trigger event" is determined. This section needs reworking into a more logical order, noting suggested major revisions.*

**Reply**: The section has been revised.
* * *
**Referee 3.27**: *Page 9 Line 9: Again, justify use of STA/LTA parameters (this could simply be that this satisfactorily isolated these signals based on*

*a visual inspection, but should specify this is the case)*

**Reply**: Justification added.
* * *
**Referee 3.28**: *Page 9 Line 22: Did you consider the wind direction? Would expect it to be pretty important!*

**Reply**: Please see our arguments regarding point 3.5.
* * *
**Referee 3.28**: *Page 10 Lines 15-16: "again, first order proxies for the susceptibility of a rockmass to thermal stress can be provided by the ambient air temperature time series and its first erivative. . . and spatially resolved sun exposure models" Citation(s) required*

**Reply**: Sentence has been modified to better link to the reference in the sentence before and making clear that we use this approach, simplified by excluding the material property terms.
* * *
**Referee 3.29**: *Page 10 Line 25: Why just March?*

**Reply**: Sentence clarified.
* * *
**Referee 3.30**: *Page 11 Line 25: ". . .adjusted to 6 and 4 respectively." . . .because. . . (presumably there was a reason)*

**Reply**: Reasons provided, now.
* * *
**Referee 3.31**: *Page 12 Line 12: Can you add this info to the figure caption too – would be useful to a reader just skimming the paper*

**Reply**: Added as suggested.
* * *
**Referee 3.32**: *Page 13 Line 2: what do you mean by "location approach frequency window"?*

**Reply**: section has been clarified.
* * *
**Referee 3.33**: *Page 14 Figure 3: Too many sub panels, so that (c) and (d) are too small to see properly. Split into multiple figures that are larger*

**Reply**: Figure has been revised. All fonts are Arial 8, panel d) has been removed and its information added to c). Figure is now designed as one panel figure, yielding a width of 8.3 cm in the typeset version of the manuscript instead of 12 cm for a two column figure. Splitting into more than one figure did not appear to be a proper alternative.
* * *
**Referee 3.34**: *Page 16 Figure 5: Label (a) and (b) with the monitoring period*

**Reply**: Labels added as suggested.
* * *
**Referee 3.35**: *Page 16 Line 5: "In contrast, most of the 32 rockfalls detected in 2015 detached and impacted in the upper and central parts. . ." How confident are you in "detachment" locations? Does detachment reliably induce detectable signals?*

**Reply**: Sentence rephrased. See also argumentation regarding point 11 of referee 1, point 3.6 of this text and the updated beginning of chapter 4.2
* * *
**Referee 3.36**: *Page 17 Figure 6: On panels (a) and (b) there is no scale for the cumulative number of events. The plotting of the time series is quite counter-intuitive. Why not plot in temporal order?*

**Reply**: Scale for cumulative number of events added. Order of a and b is guided by c. We made it now explicit by scaling a and b according to the time axis of c. This way it is possible to directly link the panels, i.e., to link weather, event distribution and elevation.
* * *
**Referee 3.37**: *Page 18 Line 6 (and again in Line 7): Should be rockfalls per month per km2*

**Reply**: Changed as suggested.
* * *
**Referee 3.38**: *Page 19 Figure 7: The panels are too small*

**Reply**: The panel is scaled to a width of 4.5 instead of 8.3 cm due to the format constraints of the discussion paper latex file. An increase of the size by 86 % should solve the issue.
* * *
**Referee 3.1**: *General comment: Do not develop a detailed discussion around lag times, although this is given a dedicated section in the results, so presumably the authors thought it to be important.*

**Reply**: In the light of what both, referee 1 and 2 demand, we actually had to provide a more detailed discussion of lag times. We also believe that this part is essential because the only possible link between triggers and rockfalls is the temporal one, i.e., the lag time analysis.
* * *
**Referee 3.39**: *Page 21 Line 30: "presumably this second rockfall was triggered by the impacts of a preceeding one" note that this is an assumption, not an observation. This is important to remember a page later (Page 25 Line 25) which states that the events would be mapped as "two discrete rockfalls" using a posterior mapping approach, suggesting that this would be incorrect. It would be inconsistent with your assumption, which may be reasonable, but you have not proven that the second rockfall was triggered by the first.*

**Reply**: Indeed, this is why we mention this case in the discussion chapter, not in the results section. Anyhow, with respect to point 13 of referee 1 we already revised this sentence to explicitly note that triggering phase five by the previous rockfall is only one possible scenario. We now also revised the second mentioned text section to state that our interpretation is a potential one.
* * *
**Referee 3.40**: *Page 22 Line 24: "no other method" this is overstating things a bit – I can immediately think of some (admittedly labour intensive and dangerous) manual methods that could collect the same information. Just need to revise the wording a little*

**Reply**: Section has been revised.
* * *
**Referee 3.50**: *Page 23 Lines 10:16: If you use all the spatial information, it looks like there are distinct patches of the cliff face active at different times of the year. These are not vertically connected so there doesn't appear to be much evidence to support your assertion that rockfall activity is actively migrating down the cliff face. Note that there is activity throughout the vertical extent of the cliff face in the 2015 data.*

**Reply**: We are not arguing for "vertically connected hotspots" in the text. In fact, we do indeed argue for "activity throughout the vertical extent of the cliff face in the 2015", but that this activity follows a vertical trend

[Figure]

Figure 1: Plot of rockfall event elevation vs. northing, sorted by date of occurrence (coloured lines starting with blue and ending with red).

with time. This trend is now also better quantified and tested in the revised version of the manuscript (see many comments of referees 1 and 2).

We interpret the situation depicted in figure 5 different. Especially in figure 5 b the colours indicative for cumulative normalised probabilities of say 4 to 8 cover the entire cliff section. The central 2015 activity hotspot (figure 5 c) stretches from just below the large ledge (event 35, about 1354 m asl.) to almost the lower part of the cliff (event 40, about 1094 m asl.). Likewise, the temporally numbered events jump between these spatial activity clusters. Thus, there is no clear trend of seasonally migrating hotspots. Obviously, activity in 2014 is concentrated at the southern section of the monitored cliff part but it does not exclusively occur there; only eight out of 17 events form this hotspot. The other half is spread over the rest of the monitored cliff.

Figure 1 illustrates this. If one follows the trajectory of rockfall occurrence through time there is never a period when rockfall activity sticks at a common area, the lines always flip for many hundreds of metres before eventually returning to a place where they have already been. The same inference could actually be made from figure 5 c when sorting the event ID number. We mentioned already in the initial manuscript version that temporally close events are always spatially separated by hundreds of meters, a point similar in consequence as what we argue for, here.

**Referee 3.51**: *Page 23 Line 20: "barely resolved" – it isn't fully resolved. Additionally for stochastic processes need several years of observations to make robust interpretations regarding seasonality*

**Reply**: The term "seasonal scale" (a temporal scale that looks at patterns and how the evolve over several months) does not imply that we argue about "seasonality" (i.e., patterns observed at a seasonal scale over many or at least several years). These explicit definitions are now added to the text. Please also see our arguments regarding point 3.3.
* * *
**Referee 3.52**: *Page 23 Line 23: ". . .there is a diurnal scale that modulates the effect of the prior one" Awkward phrasing - revise*

**Reply**: Sentence is revised.
* * *
**Referee 3.53**: *Page 23 Line 25: ". . .conditions." Citation needed*

**Reply**: We made reference to chapter 2, where all the relevant citations are given the discussion of these response times is made.
* * *
**Referee 3.54**: *Section 5.3.1: Generalisation of seasonality cannot be made based on ¡1 year of data. Should be removed*

**Reply**: Please see points 3.3 and 3.51.
* * *
**Referee 3.55**: *Section 5.3.2: "The weather-relevant scale" This seems a somewhat ambiguous term, and I'm not sure why this is separated out from the diurnal scale?*

**Reply**: It is made distinct from the diurnal scale because it acts over several days, depending on the atmospheric pressure system dynamics, and also does not include the dominant diurnal forcing mechanism: temperature changes with a 24 h period. We renamed the chapter to avoid confusions.
* * *
**Referee 3.56**: *Page 25 Line 2: slabs not "slaps"*

**Reply**: Corrected as suggested.
* * *
**Referee 3.57**: *Page 25 Figure 8: Lighter colours not easily visible (especially for freeze-thaw in panel (d)). Panels generally too small.*

**Reply**: The figure has been revised. Colours have been adjusted. The final figure size in the type set version will be 8.3 cm, not 6 cm as it is, now.
* * *
**Referee 3.58**: *Page 26 Figure 9: Rockfall activity drop is not the most intuitive of ways to express this. Much simpler just to plot the cumulative number of events. Also, use a legend and only one vertical axis – this would be much clearer.*

**Reply**: The figure and its discussion has been removed from the manuscript, please see comment 27 of referee 2.
* * *
**Referee 3.59**: *Section 5.3.4. For reference, it would be good to know approximately what gravitational acceleration would be required before there is likely to be significant risk of moving rocks/crack propagation.*

**Reply**: References and their discussion have been added.
* * *
**Referee 3.60**: *Section 5.4 I'm not sure that this section can be justified based on comment in major revisions*

**Reply**: Considering the major revisions of the justifying sections for this chapter (i.e., 4.2 and 5.2) and in the light to our replies to point 3.50 and the comments of referee 1 and 2 we believe the vertical trend of rockfall activity is now properly justified and needs to be explained by this section 5.4.

**1    Cited literatue**

Dietze, M., Mohadjer, S., Turowski, J. M., Ehlers, T. A., and Hovius, N.: Validity, precision and limitations of seismic rockfall monitoring, Earth Surf. Dynam. Discuss., in review, 2017 a.

Larose, E., Carrire, S., Voisin, C., Bottelin, P., Baillet, L., Gueguen, P., Walter, F., Jongmans, D., Guillier, B., Garambois, S., Gimbert, F., and Massey, C., Environmental Seismology: What can we learn on Earth surface processes with ambient noise ?, Journal of Applied Geophysics (2015), doi: 10.1016/j.jappgeo.2015.02.001. 2015.

Strunden, J., Ehlers, T. A., Brehm, D., and Nettesheim, M.: Spatial and temporal variations in rockfall determined from TLS measurements in a deglaciated valley, Switzerland, Journal of Geophysical Research, 120, 123, doi:10.1002/2014JF003274, 2014.

---

## Author Response (AR2)

**Response to referee comments – referee 1**

[Spatiotemporal patterns, triggers and anatomies of seismically detected rockfalls]

September 29, 2017

We would like to thank the editor for the encouraging and helpful comments, all of them obviously devoted to improve the quality and impact of the manuscript.

All suggested typo or phrasing revisions were implemented as suggested.
* * *
**Editor 1**: *Consider breaking into two sentences.*

**Reply**: Done as suggested ("Seismic data allows the classification of rockfall activity into two distinct phenomenological types. The signals can be used to discern multiple rock mass releases from the same spot, identify...")
* * *
**Editor 1**: *Consider adding a sentence here explaining how you know that freeze-thaw transitions caused these rockfalls. Is this from a temperature record?*

**Reply**: Sentence added ("Freeze-thaw-transitions, approximated at first order from air temperature time series, account...")
* * *
**Editor 1**: *Say why (maybe this is obvious, but I think it is useful to highlight that seismic techniques can deliver almost continuous temporal coverage.*

**Reply**: Done as suggested ("The temporal information delivered by these methods is not very precise as it is bound to the survey lapse times, which are typically on the order of weeks to years." and "They allow precise temporal fixes of rockfall event initiation and duration because they record continous high resolution signals of geomorphic activity.")
* * *
**Editor 1**: *Awkward phrase. Rewrite.*

**Reply**: Done as suggested ("Combining spatial and temporal rockfall patterns, we identify a rockfall activity zone that consistently shifts down the cliff over the course of the season, and quantify the effect of diurnal forcing on event activity within the composed catalogue.")
* * *
**Editor 1**: *Sureley there are dozens of papers that do this, many predating this paper? I would add some references here.*

**Reply**: Additional key references added.
* * *
**Editor 1**: *A suggestion: this section contains a list of possible triggers and each trigger section suggests possible "fingerprints" of these different trigger mechanisms. I think it might be useful if the authors prepared a table that gave the different mechanisms and listed the signals within the seismic data that could be used to differentiate these mechanisms. The authors are much better placed than I to determine if this would be useful but I think the authors should consider such a table (I could image something like that appearing in many talks and book chapters in the future).*

**Reply**: A summarising table is added now and also contains short descriptions and key references for a potential seismic approach to asses trigger actitivity. Due to the Latex manuscript template it was not possible to have a single table that fits onto one page. Thus, the final layout shall have a minor modification and merge the two tables of the current manuscript version.
* * *
**Editor 1**: *Add a few words of explanation. There is a ratio. A ratio of what to what? The ground velocity vs the noise in the background velocity?*

**Reply**: Done as suggested ("This algorithm is sensitive to instantaneous rises in the recorded seismic signals, which affect the long-term running average only marginally while raising the short-term running average severely and thus increase the ratio at the onset of seismic activity.")
* * *
**Editor 1**: *If you are just using random names for these stations, surely you can come up with better ones. You next installation should be called "Xandor, Lord of the Three Eyed Conger Eels"*

**Reply**: Well, the names are not random, though profoundly explaining their origin shall remain the first author's well hidden secret or may be a matter of a personal conversation. If the editor comes up with an interesting research project in the North of the United Kingdom, the first author would be chuffed to name one of the seismic stations "Xandor, Lord of the Three Eyed Conger Eels".
* * *
**Editor 1**: *If you want to keep the reference to the triangular shape you can add a sentence here saying this looks somewhat triangular on the seismogram*

*But I don't really see why you need that beyond the description of what is happening to the frequencies as they evolve in time.*

**Reply**: Sentence part with triagular shape has been removed to be consistent as suggested.
* * *
**Editor 1***: say why (calibrated somehow in the 2017 paper?*

**Reply**: Additional information provided, now ("velocity was set to 2700 m s$^{-1}$, which provided the best location accuracy in this study area (Dietze et al., 2017).").
* * *
**Editor 1***: This is a little worrying in that every time I have used MCMC methods I've needed chains much longer than 1000 links to get a stable behaviour (i.e., the burn in period is longer than 1000 iterations). Did you test to see if changing the length of the chain gave you different answers?*

**Reply**: We had tested this point during the data analysis period and added information to the manuscript ("Initial tests showed that stable, reproducable plots emerge with already 500 MCM runs and larger chains did not improve the quality of the results"). For this applicartion we just had to combine two independent variables, time of event and kernel bandwidth, so there is not a large number of possible combinations that would require long chains. Please also see figure 1 for a short test on the effect of changing the chain length for 50 randomly distributed values, based on the following R code:

```
**define data set**
x <- runif(50)

**perform MCM-based resamling and density estimate approach 101 times**
d <- lapply(X = 1:101, FUN = function(i, x) {

  x_i <- sample(x = x,
                size = round(length(x) * runif(n = 1,
                                               min = 0.8,
                                               max = 1),
                             digits = 0),
                replace = FALSE)

  k <- density(x = x_i,
               bw = 0.1,
```

[Figure]

Figure 1: MCMC-based kernel density estimates for 101, 1001, and 10001 MC runs, each resampling the same series of 50 values with a resampling size between 80 and 100 %. The plots depict mean ± 1 standard deviation and show virtually no difference, suggesting that for such data also MCMC chain lengths of about 101 are sufficient to generate stable results.

```
                    n = 1000,
                    from = -2,
                    to = 2)$y

}, x)

**convert list to matrix and calculate columnwise mean and sd**
d <- as.matrix(do.call(rbind, d))
d_m <- apply(X = d, MARGIN = 2, FUN = mean)
d_s <- apply(X = d, MARGIN = 2, FUN = sd)

plot(NA,
     xlim = c(-2, 2),
     ylim = c(0, max(d_m + d_s)),
     main = "n = 101")
lines(x = seq(-2, 2, length.out = 1000),
      y = d_m)
lines(x = seq(-2, 2, length.out = 1000),
      y = d_m - d_s,
      col = "grey")
lines(x = seq(-2, 2, length.out = 1000),
      y = d_m + d_s,
      col = "grey")

dev.off()
```

**Editor 1**: *Sweaty herbs is a good name for a station.*

**Reply**: Thanks, we also consider this a good name.
* * *
**Editor 1**: *Independently calculated from what data? Refer to either figure or TLS data.*

**Reply**: Done as suggested (”This implies an TLS-based free fall distance of not more than 106 m (figure 3 d)“).
* * *
**Editor 1**: *Check figure formatting. Hard to see anything here.*

**Reply**: The figure is scaled to 69 % of its actual size due to the manuscript Latex style. In the final version of the article this issue should be solved.
* * *
**Editor 1**: *This caption needs an explanation of what the MCMC fit line is doing here.*

**Reply**: Added as suggested (”Semi-transparent graphs are Monte-Carlo based exponential fits of the elevations of the events denoted in grey colour“).
* * *
**Editor 1**: *I'm a bit uncomfortable with this. Is it really needed in the context of the paper? Based on the figures, there are a range of starting rockfall locations and these seem to get lower as time progressed during the monitoring. But at any given point there is quite a range in the initiation points. So really what one has (or this is how I interpret the data) a moving window there there was rockfall activity and the median elevation of this activity moved lower through time. I feel like presenting this equation could lead to abuse by others where a model simulated a ”buzz saw” acting on a particular elevation through time. The authors might consider either some lines of caution in using this equation with particular reference to the probabilistic nature of the initiation elevations.*

**Reply**: The inclusion of this fit is mainly a result of the initial reviewer comments. We now added a sentence that highlights the significant scatter of the detachment elevations around the overall trend (”...root mean square error of 76 m. Thus, there is significant scatter in this overall trend, underlining that the model only describes a first order effect visible in the data, which is modulated by further factors of influence that impose a strong stochastic effect“).
* * *
**Editor 1***: section 2?*

**Reply**: Changed as suggested.